# Complex 33-beam simulated galactic cosmic radiation exposure impacts cognitive function and prefrontal cortex neurotransmitter networks in male mice

Rajeev I. Desai [1,2,3] ✉, Brian D. Kangas [1,2], Oanh T. Luc[1,2], Eleana Solakidou[3,4], Evan C. Smith[3], Monica H. Dawes [1,2], Xiaoyu Ma[3], Alexandros Makriyannis[3], Subhamoy Chatterjee[5], Maher A. Dayeh[6,7], Andrés Muñoz-Jaramillo [5], Mihir I. Desai[6,7] & Charles L. Limoli[8]

Astronauts will encounter extended exposure to galactic cosmic radiation (GCR) during deep space exploration, which could impair brain function. Here, we report that in male mice, acute or chronic GCR exposure did not modify reward sensitivity but did adversely affect attentional processes and increased reaction times. Potassium (K+)-stimulation in the prefrontal cortex (PFC) elevated dopamine (DA) but abolished temporal DA responsiveness after acute and chronic GCR exposure. Unlike acute GCR, chronic GCR increased levels of all other neurotransmitters, with differences evident between groups after higher K+-stimulation. Correlational and machine learning analysis showed that acute and chronic GCR exposure differentially reorganized the connection strength and causation of DA and other PFC neurotransmitter networks compared to controls which may explain space radiation-induced neurocognitive deficits.

Human deep space exploration to the Moon and Mars will require travel beyond the Earth's protective magnetic field and involve significant health challenges for astronauts. This includes extended exposure to Galactic Cosmic Radiation (GCR) consisting of particles of high atomic number, energy, and charge (HZE). Due to shielding constraints, charged particles penetrate the spacecraft rendering astronauts unprotected against GCR exposure. During interplanetary travel, each cell within an astronaut's body is likely to be hit by either a proton or secondary particle every few days and by a HZE ion approximately once a month[1,2]. Mounting evidence suggests that exposure to space radiation can cause variable damage to the central nervous system (CNS)[3-8]. Estimates indicate that during a Mars

mission, nearly every neural cell will be transversed once every three days[3-8] by various combinations of protons and/or HZE nuclei. These estimates have raised concerns that continuous exposure to GCR may damage CNS functionality and jeopardize mission success[5-11].

Investigations into how extended GCR exposure adversely affects the brain have been impeded by the limited number of humans that have traveled in deep space. This fact has necessitated ground-based studies in which laboratory animals are exposed to different simulated patterns of space radiation to probe the multitude of CNS effects. Early CNS studies in rodent models primarily focused on acute effects following simplified single ion exposures that were not accurate representations of the low dose and low dose rate deep space radiation

[1]Department of Psychiatry, Harvard Medical School, Boston, MA 02115, USA. [2]Behavioral Biology Program, McLean Hospital, Belmont, MA 02478, USA. [3]Center for Drug Discovery, Department of Pharmaceutical Sciences, Northeastern University, Boston, MA 02115, USA. [4]Medical School, University of Crete, Heraklion, Greece. [5]Southwest Research Institute, Boulder, CO 80302, USA. [6]Southwest Research Institute, San Antonio, TX 78238, USA. [7]University of San Antonio, San Antonio, TX 78249, USA. [8]Department of Radiation Oncology, University of California, Irvine, Orange, CA 92697, USA. ✉e-mail: rdesai@mclean.harvard.edu

environment in which astronauts are continuously subjected[5–11]. More recent results obtained with advancements in beam delivery technology have indicated that space-relevant doses and dose rates of various single-ion, mixed-field GCR, and neutron exposures lead to cellular and molecular damage to the CNS that likely results in short- and long-term decrements in neurocognitive performance[5–26].

Although the precise mechanisms through which GCR produces damaging effects on the brain remains unknown, past studies have linked behavioral decrements to compromised structural plasticity and network level connectivity, as well as elevated microgliosis, astrogliosis, and inflammatory signaling in the irradiated brain[5–26]. In nearly every instance, these studies found poor correlation with the particle linear energy transfer (LET: keV/μm) or dose-responsiveness when realistic exposures (≤50 cGy) were evaluated[5–26]. Investigators have also attempted to relate neurocognitive deficits to the neurochemical sequelae of high doses of single-ion HZE particle and proton radiation. However, assay limitations and a focus on the neurotransmitter dopamine (DA) and resultant behavioral effects have yielded mixed results, i.e., DA activity related to some but not all behavioral deficits[8,13–15]. More recently, studies have reported that space radiation exposure also produces persistent deleterious changes in other neurotransmitters involved in cognition[27] such as glutamate (Glu), γ-aminobutyric acid (GABA), and serotonin (5-HT)[17–20,28,29] as well as monoamine metabolism[30].

The recent availability of complex 33-beam terrestrial GCR simulation, advances in rodent touchscreen technology, and development of in vivo microdialysis combined with biochemical techniques using liquid chromatography mass spectrometry (LC-MS/MS), offers a unique opportunity to delineate how space-relevant GCR exposure alters in vivo neurotransmitter function to impact neurocognitive performance. Some recent studies utilized various touchscreen-based neurocognitive assays in rodents to find that exposure to single-ion or 6-mixed-ion beam either improved, impaired, or did not alter performance[31–33]. Likewise, we found that acute or chronic delivery of the 33-beam GCR exposure produced less severe neurocognitive and electrophysiological impairments in mice, suggesting that the impact of space-relevant radiation exposure may be less detrimental than simplified irradiation paradigms[34–36]. Collectively, such data highlight the critical need to study the effects of more space-relevant radiation exposures on multiple within-subject neurocognitive and neurotransmitter endpoints

to better estimate the risk of space radiation exposure to CNS functionality.

Given that brain neurotransmitters provide informative readouts of neurocognitive function, it is critical to delineate how exposure to space-relevant radiation alters neurotransmitter activity in vivo to impact neurocognitive performance. Here, we conducted studies in adult mice to examine the impact of the complex 33-beam terrestrial GCR simulation on prefrontal cortex (PFC) neurotransmitters involved in translational touchscreen-based complex behavior. Based on well-established reports, the PFC was selected because of the extensive body of evidence showing its involvement in higher order behaviors and cognitive function, including motivation and psychomotor vigilance that are likely required for mission success during spaceflight[37–41]. With regard to neurotransmitters, we primarily focused on documenting changes in monoamines, Glu, and GABA as a group of key neurotransmitters in the PFC because prior work has shown: a) that they all play an important role in various cognitive processes, including motivation and attention and b) that their dysregulation is directly linked to a variety of neuropsychiatric conditions[27,37–44]. We report findings describing how the PFC neurotransmitter networks are reorganized by realistic complex GCR simulations and how that informs on neurocognitive outcomes to provide a foundational and comprehensive outlook regarding the impact of space radiation on CNS function.

## Results

Figure 1 shows a detailed timeline of experimental protocols. Mice received a 33-beam GCR simulation for a total dose of 40 cGy, either all-at-once (acute) or 49.9 cGy over 4 weeks (chronic). Next, economic demand functions for a palatable food reinforcer were determined to assess changes in reward sensitivity. Thereafter, sustained attention was examined during a psychomotor vigilance task. The same mice were then probed by combined in vivo microdialysis with LC-MS/MS in order to quantify potassium (K+)-evoked (i.e., depolarization elicited) changes in multiple neurotransmitters in the PFC. Finally, a combination of statistical and machine learning techniques (correlation analysis, Granger causality test, and iterative random forests) was used to determine how acute and chronic GCR altered PFC neurotransmitter networks to impact neurocognitive performance.

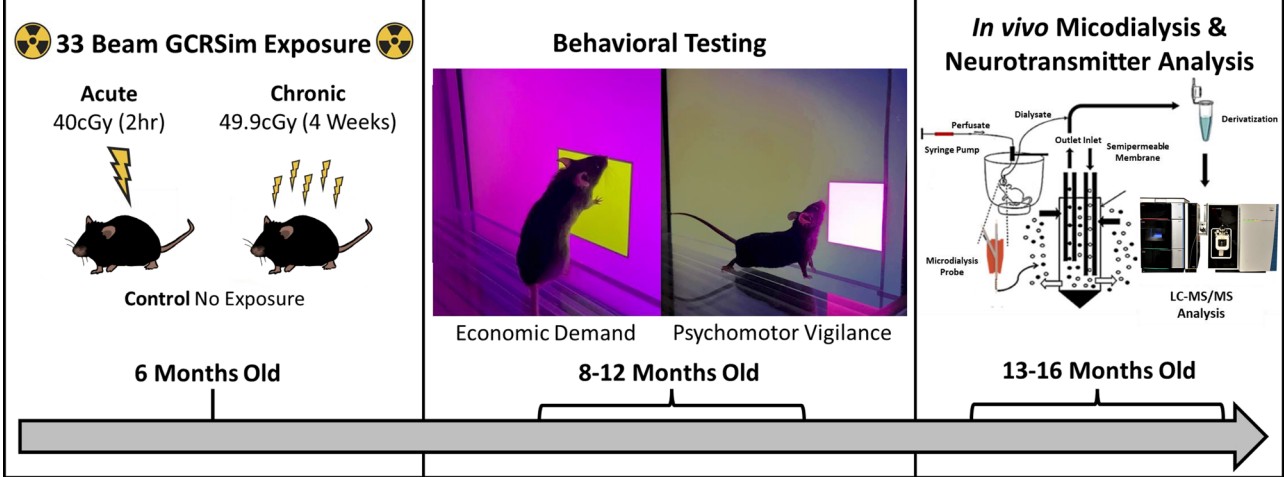

**Fig. 1 | Timeline of experimental protocols in mice.** All subjects were shipped directly to NASA Space Radiation Laboratory, Brookhaven National Laboratory (Long Island, NY) for exposure to galactic cosmic radiation simulation (GCRSim). Following acute or chronic exposure to GCRSim and post-exposure recovery for 3–5 days, all subjects (including controls) were shipped to McLean Hospital, Harvard Medical School (Belmont, MA). After an 8-week mandatory quarantine period, all subjects were interrogated for changes in neurocognitive performance using touchscreen-based assay and neurotransmitter function using in vivo microdialysis/liquid chromatography mass spectrometry (LC-MS/MS) analysis over a 7-month period.

## Effects of acute or chronic GCR exposure on motivation and sustained vigilance

**Economic demand.** An important goal for evaluating the impact of GCR on cognition is to employ tasks in laboratory animals that meaningfully translate to humans. Thus, we first investigated the impact of acute or chronic GCR exposure in mice engaged in a touchscreen-based economic demand task designed to measure motivation (Fig. 2a). In this assay, subjects respond to a geometric stimulus on the touchscreen under escalating response requirements (price) for small amounts of a highly palatable liquid reward (commodity) to examine the extent to which subjects will defend consumption in the face of increasing cost (Fig. 2a; Online Methods). As shown in Fig. 2b, the exponential demand equation provided a strong fit to the findings of decreased consumption as a function of increased price in all three treatment groups ($R^2 = 0.98-0.99$ [control], 0.97–1.0 [acute], 0.94–0.99 [chronic]). A milk concentration-related rank order of consumption was apparent in the control and acute treatment groups. Accordingly, for these treatment groups, consumption of 50% milk was associated with the most inelastic curve relative to other milk concentrations, followed by 20%, 5%, and 0%. The chronic treatment group generally followed this rank order, with the exception of responding for the 20% milk concentration, which was comparatively more inelastic at higher response requirements than 50%. Moreover, Fig. 2c shows that although there was an orderly positive relationship between milk concentration and both $Q_O$ and essential value, there were no systematic differences across treatment groups. These outcomes are confirmed statistically by a significant interaction for milk concentration ($Q_O$: $F[3,84] = 54.74$,

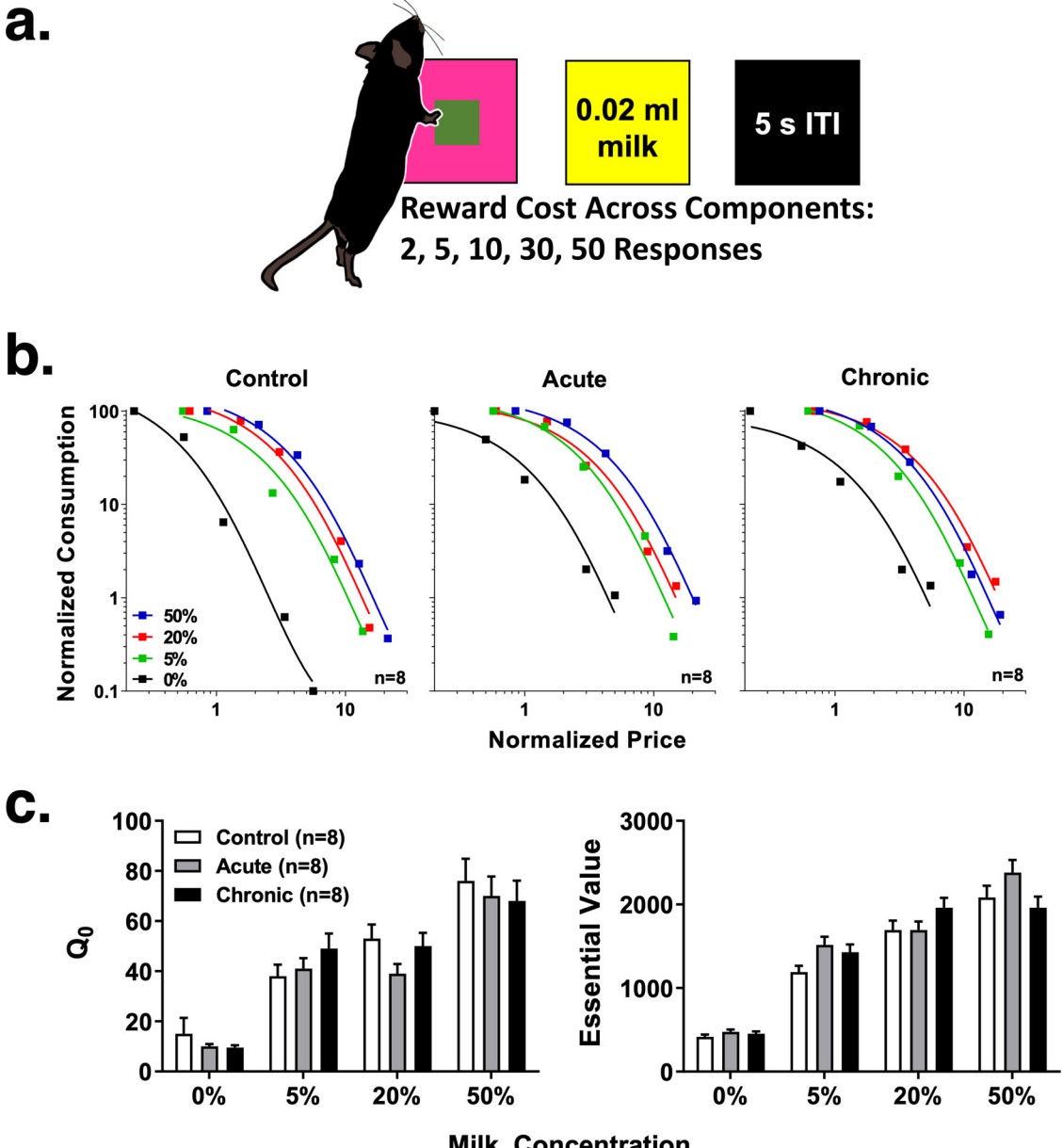

**Fig. 2 | Acute and chronic GCR fail to impact reward sensitivity or motivation in an economic demand task. a** Task schematic. **b** Normalized demand functions for sweetened condensed milk concentrations for the control (left panel), acute (middle panel), and chronic (right panel) treatment groups ($n = 8$/group). Normalized consumption is expressed as a percentage of $Q_O$ (demand intensity at price 0) and price is expressed as the number of responses required to produce 1% of $Q_O$. **c** $Q_O$ (left panel) and essential value (right panel) for the control (white bars), acute (gray bars), and chronic (black bars) treatment groups (± standard error) across sweetened condensed milk concentrations. To evaluate the statistical significance of GCR exposure on economic demand parameters, data were subjected to two-way analysis of variance (ANOVA). ITI intertrial interval.

$p < 0.0001$; essential value: $F[3,84] = 155.90$, $p < 0.0001$), but not for the treatment group ($Q_0$: $F[2,84] = 0.99$, $p = 0.38$; essential value: $F[2,84] = 2.89$, $p = 0.06$).

**Psychomotor vigilance.** Next, we then determined whether these same mice would display any deficits on sustained attention using the psychomotor vigilance task. In this assay, subjects were required to carefully attend and respond effectively to intermittent geometric stimuli presented briefly in variable locations and after variable intervals on the touchscreen to obtain the palatable liquid reward (Fig. 3a; Online Methods). Figure 3b presents the mean titrated reaction time for each treatment group across the 15-day testing condition. Significant differences in task performance across treatment groups were especially prominent during the first week of testing, with the control group quickly approaching the asymptotic level in mean titrated reaction times of approximately 2.5 s. Both the acute and chronic groups also displayed quicker titrated reaction times across the 15-day testing condition, but this developed at a slower rate, which was confirmed by a significant main effect of treatment ($F[2,315] = 30.56$, $p < 0.0001$). Moreover, although differences in reaction times among treatment groups did not reach statistical criteria after day 6, acute and chronic group mean titrated reaction times failed to fully approximate levels observed in the control group by the end of 15-day testing condition, with both GCR treatment groups having reaction times plateau at values approximately 1 s longer than the average control subject.

**Effects of acute or chronic GCR exposure on neurotransmitters in the PFC**

**Baseline and high K+-evoked extracellular levels of neurotransmitters in the PFC.** In the same mice, we investigated how GCR impacts neurotransmitters in the PFC–a key brain region implicated in reward sensitivity and attention[37–41,43]. Using microdialysis probes (Fig. 4a, b) targeting the PFC (Fig. 4b, c) and LC-MS/MS techniques (Fig. 4d, e), we determined real-time baseline (4 mM K+) and high K+ (30, 60, and 120 mM)-evoked changes in multiple neurotransmitters (Fig. 4a–e; Online Methods). See Supplementary Table 1 for cumulative basal and high K+-evoked neurotransmitter levels in each group. We found that levels of DA and 5-HT were significantly higher in acute [2.71- and 1.75-fold, respectively] and chronic [2.14- and 1.96, respectively] groups compared the controls (Fig. 5a, b, respectively). Baseline levels of norepinephrine (NE) were also significantly greater (1.60-fold) in the chronic group compared to controls (Fig. 5c). In contrast, neither acute or chronic GCR significantly changed baseline levels of Glu and GABA compared to control values (Supplementary Table 1, Fig. 5d, e, respectively). A significant difference between levels of NE and Glu in the acute vs chronic GCR groups was observed (2.19-fold and 1.45-fold, Fig. 5c, d, respectively); for the other neurotransmitters, the basal levels in acute vs chronic GCR groups were unchanged (Fig. 5a–e).

Increasing K+-stimulation uncovered distinct differences in neurotransmitters among the treatment groups. DA (1.87–2.41-fold) and 5-HT (1.78–1.86-fold) were significantly higher in the acute group compared to controls during two or more higher K+ concentrations (Fig. 5a, b, respectively), whereas other neurotransmitters did not significantly differ between acute and controls (Fig. 5c–e, respectively). In contrast, 5-HT (1.87–2.19-fold), NE (1.34–1.40-fold), Glu (1.45–1.65-fold), and GABA (1.50–1.92-fold) levels were significantly higher in the chronic group compared to controls during stimulation with two or more higher K+ concentrations (Fig. 5b–e, respectively). DA levels in the chronic group were generally higher compared to controls with significant differences observed at 120 mM K+ (1.69-fold; Fig. 5a). While NE (60 mM K+: 1.45-fold), Glu (120 mM K+: 1.56-fold), and GABA (120 mM K+: 1.62-fold) differed between the two GCR groups (Fig. 5d, e, respectively), the other neurotransmitters were unchanged (Fig. 5a–e).

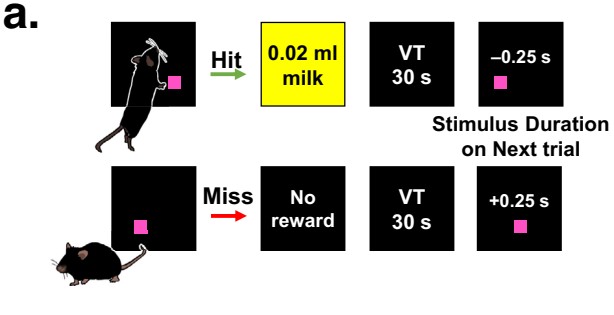

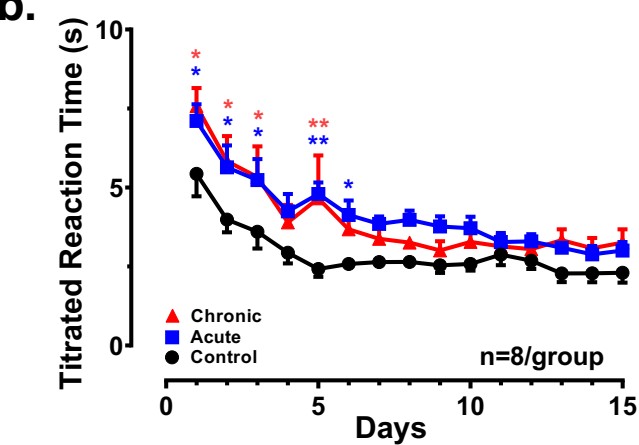

**Fig. 3 | Acute and chronic GCR produce attentional deficits in a psychomotor vigilance task. a** Task Schematic. **b** Mean (±SEM) titrated reaction times (s) for the control (black data series), acute (blue data series), and chronic (red data series) treatment groups ($n = 8$/group) across the 15-session condition. A two-way ANOVA followed by a *post-hoc* Tukey test when appropriate was used. *$p < 0.05$; **$p < 0.01$ represents significant difference compared to controls. VT variable time.

**Time course of neurotransmitter responsiveness in the PFC.** In control mice, 30 mM induced a rapid increase in DA (-394 ± 122% basal nM levels) and NE (-157 ± 37% basal nM levels) within 20-min that rapidly returned to near baseline values (40–80 min; Fig. 6a, c, respectively). Increasing the K+ to 60- or 120-mM also elevated DA (-259–293 ± 98–102% basal nM levels) and NE (207–221 ± 31–57% basal nM levels) about 40-min after each K+ stimulation with levels of both neurotransmitters returning to baseline values about 20–40 min after peak effects (Fig. 6a, c, respectively). In contrast, acute or chronic GCR exposure completely abolished normal rapid elevations and decreases in DA responsiveness during each increasing K+ stimulation (Fig. 6a). DA levels in both GCR groups were generally lower than controls throughout the experiment (Fig. 6a). Further analysis confirmed that DA levels in the control group were significantly greater than basal values and levels in the acute and chronic GCR groups at 20 min during 30 mM (Fig. 6a; -394 ± 122 vs 116 or 94 ± 45 or 39% basal nM levels, respectively; $p$ values < 0.05). Unlike DA, acute GCR mice showed normal K+-stimulated rapid increases and decline in NE responsiveness (Fig. 6c). NE changes in chronic group were generally lower but were not significantly different compared to controls (Fig. 6c). Levels of 5-HT were not consistently altered in all groups during 30- or 60-mM, but a steady rise in 5-HT in all three groups was observed at 120 mM; except for some separation between acute and control groups at 240 min ($p < 0.05$) no differences were observed (Fig. 6b: $p$ values > 0.05).

In control mice, 30- and 60-mM decreased Glu levels (Fig. 6d: 120–160 min, -53 ±9.1% basal nM levels; $p$ values < 0.05), whereas GABA was unaffected (Fig. 6e). Increasing the K+ to 120 mM restored Glu levels towards basal values (Fig. 6d), but GABA was significantly

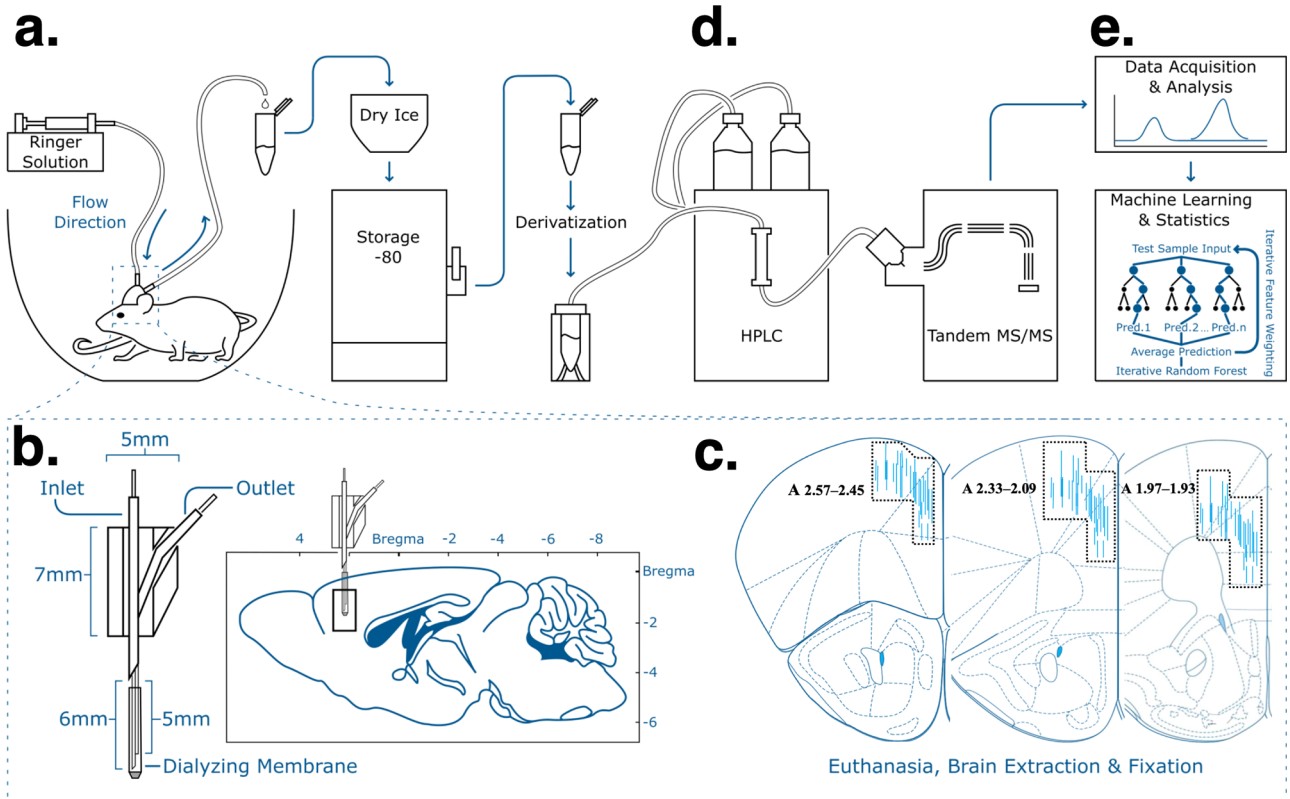

**Fig. 4 | Flowchart illustration of in vivo microdialysis experimental set-up, design, sample collection, sample analysis, and histology.** Samples were collected from control and GCR exposed mice using custom-designed microdialysis probes as shown in enlargement schematic (**a**; see online methods for details of probe construction) with the dialysis membrane targeting the PFC as shown in the enlargement schematic (**b**). A CMA/102 microdialysis syringe pump was used to perfuse the dialysis probes with Ringer solution consisting of different concentrations of K+. Dialysate samples were collected in microcentrifuge tubes placed in dry ice and samples were stored at −80 °C until quantification. **c** After completion of microdialysis experiments, all subjects were euthanized and brains were removed, fixed, and sectioned. Drawings of the forebrain sections based on

Paxinos and Franklin[64] with superimposed rectangles that show the confines within which the microdialysis probe tracks were considered to be in the PFC. Data were only included from subjects with probe tracks within the rectangles and the anterior coordinate (measured from bregma) is located in each section. **d** Dialysate samples from mice in which the probes were located in the PFC were quantified to determine levels of neurotransmitters using high-performance liquid chromatography (HPLC) coupled with tandem mass spectrometry (MS/MS). Neurotransmitters in dialysate samples were analyzed using standard statistical tools, as well as machine learning approaches to determine changes in the PFC neurochemical network (**e**).

elevated above basal values in control mice (Fig. 6e; 200–240 min: ~515 ± 156% basal nM levels; $p$ values < 0.05). In acute and chronic groups, 30- and 60-mM did not change Glu or GABA levels compared to controls (Fig. 6d–e, respectively). However, 120 mM K+ increased Glu in the chronic group compared to controls and acute GCR mice (200–220 min: ~134 ± 33% basal nM levels; $p$ values < 0.05); controls and acute GCR mice produced similar effects on Glu (Fig. 6d). Compared to the controls, GABA levels were significantly lower (200 min: ~223, ±60% basal nM levels; $p$ values < 0.05) and higher (220 min: ~957, ±283% basal nM levels; $p$ values < 0.05) in, respectively, acute and chronic GCR mice during 120 mM (Fig. 6e). The increased GABA levels from 200 to 240 min in chronic mice were also significantly different than in the acute group ($p$ < 0.05; Fig. 6e).

**Reorganization of neurotransmitter network in the PFC.** We used Pearson correlation analysis to determine the relationships among neurotransmitters (Supplemental Tables S1–S4). Under baseline 4 mM, correlations between neurotransmitters become more positive and negative after acute and chronic GCR exposure, respectively, relative to controls (Fig. 7a). Increasing K+ from 30- to 120-mM produced small increases in positive correlations between the neurotransmitters in controls, whereas correlations between neurotransmitters in the GCR groups compared to 4 mM and controls at higher K+ concentrations were significantly higher and more positive (Fig. 7a).

We plotted heat maps of pairwise neurotransmitter correlations (Fig. 7b) and constructed corresponding neurotransmitter networks based on values that only include Pearson's correlation ≥ 0.5 with a significance $p$-value of <0.05 (Fig. 7c; Supplemental Tables S1–S4). During 4 mM, there was a significant and strong positive correlation between Glu-GABA in controls (Fig. 7b, c). In control mice, this Glu-GABA network was no longer linked at 30 mM and 60 mM but was preserved at 120 mM with a significant and positive correlation between the two neurotransmitters (Fig. 7b, c). In the acute group, these neurotransmitter interactions were altered during 4 mM baseline conditions; the Glu-GABA interaction was not significant and no longer correlated (Fig. 7b, c). In acute GCR mice stimulated with 30 mM, there were significant and strong positive correlations between three neurotransmitter pairs (DA-NE, NE-Glu, and Glu-GABA) that were not present in control mice at 30 mM (Fig. 7b, c). Glu and GABA were also significantly and positively correlated, and no interactions were observed between GABA and monoamines (Fig. 7b, c). 60 mM also resulted in significant and positive correlations between two neurotransmitter pairs in the acute GCR mice (DA-NE and NE-GABA). However, this differed from 30 mM, in that no neurotransmitter correlated with Glu; instead DA was positively correlated with NE and NE directly interacted with GABA (Fig. 7b, c). Except for Glu, 120 mM induced significant and strong positive correlations between all neurotransmitters resulting in multiple connecting lines for DA, other

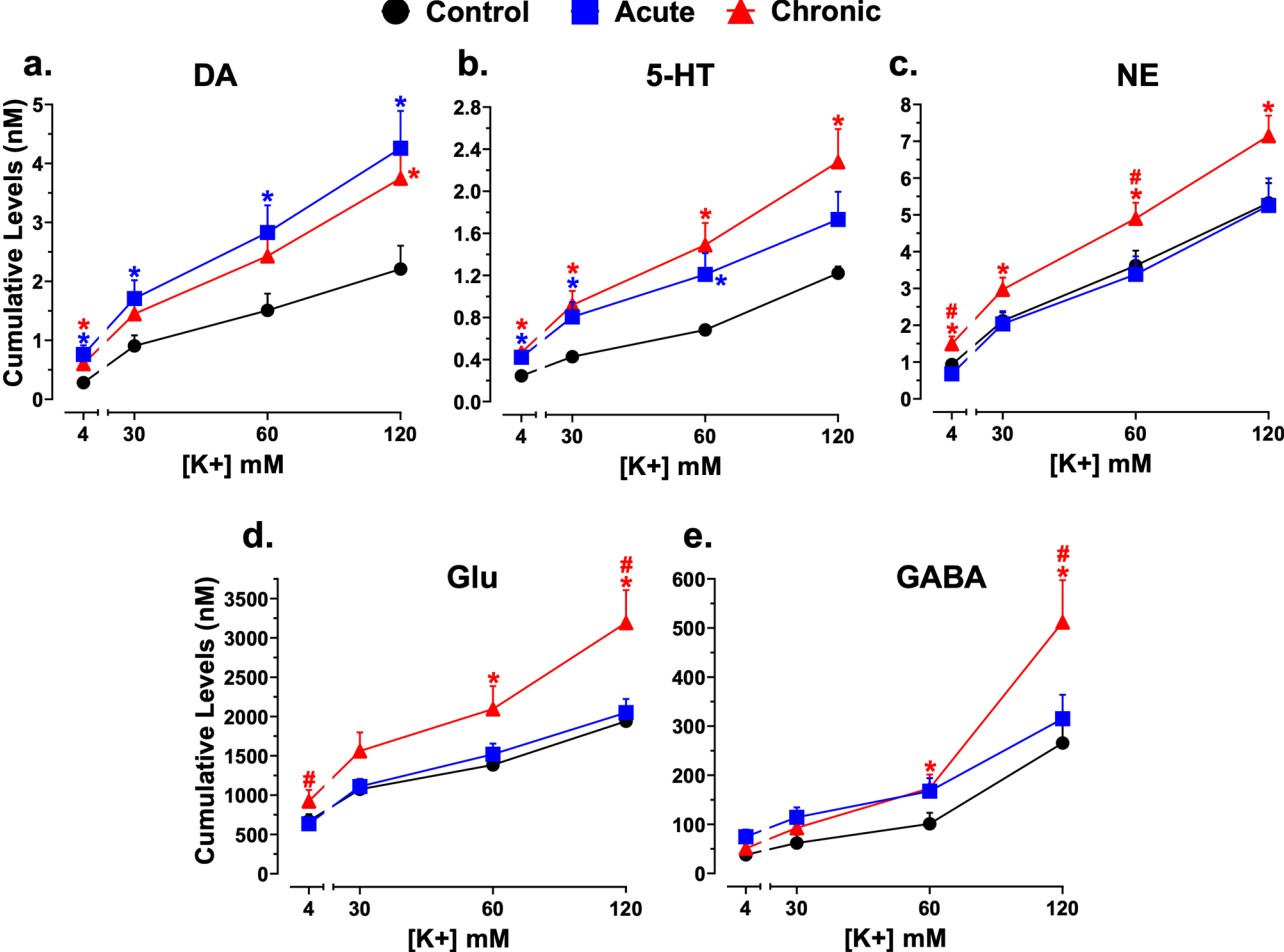

**Fig. 5 | Acute and chronic GCR exposure produced differences in K+-evoked neurotransmitter levels in the PFC.** Cumulative levels of DA (**a**), 5-HT (**b**), NE (**c**), Glu (**d**), or GABA (**e**) at each K+ concentration in the PFC of control (*n* = 10), acute (*n* = 10), or chronic (*n* = 8) GCR exposed mice are shown. Ordinates, cumulative levels of DA, 5-HT, NE, Glu, or GABA in nM; abscissae, K+ concentration in mM. Each data point represents the mean (±S.E.M.) nM concentrations of the cumulative levels of all four dialysate samples taken at 4-, 30-, 60-, or 120-mM K+ concentration. An unpaired two-tailed *t* test was used to determine differences in cumulative increases in neurotransmitter levels among groups. *$P < 0.05$ represents significant difference compared to controls within each concentration. #$P < 0.05$ represents significant differences between acute and chronic GCR treated mice at each concentration. See Supplementary Table 1 for further details on statistics. DA dopamine, 5-HT serotonin, NE norepinephrine, Glu glutamate, GABA γ-aminobutyric acid.

monoamines, and GABA in the acute GCR group (i.e., DA–5-HT/NE/GABA, GABA–5-HT/NE Fig. 7b, c). After chronic GCR, the Glu–GABA interaction was preserved during 4 mM and a significant and positive interaction was observed between DA–NE (Fig. 7a–c). In general, increasing K+ resulted in distinct neurotransmitter interactions in the chronic GCR group compared to others. 30 mM produced significant positive correlations between DA and 5-HT/NE and retained the Glu–GABA networks that were evident in control mice during 4 mM, but not at 30 mM (Fig. 7b, c). Unlike control and acute groups, significant positive correlations were only observed between certain neurotransmitters in the chronic group during 60 (DA–5-HT and Glu–GABA) and 120 (DA–5-HT, 5-HT–GABA, and Glu–GABA) mM K+ (Fig. 7b, c).

**Machine learning prediction models to determine PFC neurotransmitter networks.** Simplified correlation-based neurotransmitter networks that rely on Pearson correlation do not evaluate either nonlinear relationships or causality between neurotransmitter levels for different target classes. Thus, we used an iterative random forest (iRF) based regression model called iRF-LOOP to predict how acute or chronic GCR exposure alters PFC neurotransmitter interactions compared to controls (Fig. 8). This analysis revealed a highly complex and interconnected neurotransmitter network that changes with increasing K+-stimulation across treatment groups (Fig. 8a, b). Overall, during each K+ stimulation, GABA strongly predicted Glu levels but not necessarily the reverse. This connection strength was generally not impacted after acute or chronic GCR compared to controls (Fig. 8a, b). While Glu predicted levels of 5-HT/NE and GABA predicted levels of DA/5-HT/NE during stimulation with one or more K+ concentrations in control mice, these interactions were reorganized in the acute GCR group. Evidence of this is apparent by the elimination of the GABA–DA connection, a reduction (except for 120 mM) of the GABA–5-HT interaction, the emergence of a stronger DA–NE or DA–Glu relationship, where DA predicted levels of NE and vice versa or DA predicted levels of Glu (Fig. 8a, b). Likewise, a reduction in interactions between GABA and DA/5-HT/NE were observed in the chronic group compared to controls. In general, the interactions between Glu and monoamines did not significantly alter compared to the control group except for some small variations in connection strength and direction during one or more K+ concentration (Fig. 8a, b). Unlike the acute group, no DA–Glu connections were found in chronic GCR mice (Fig. 8a, b). Strong interactions were also predicted between NE and DA/5-HT and vice versa in control mice during stimulation with one or more K+ concentrations, but DA–5-HT were not connected in these mice except

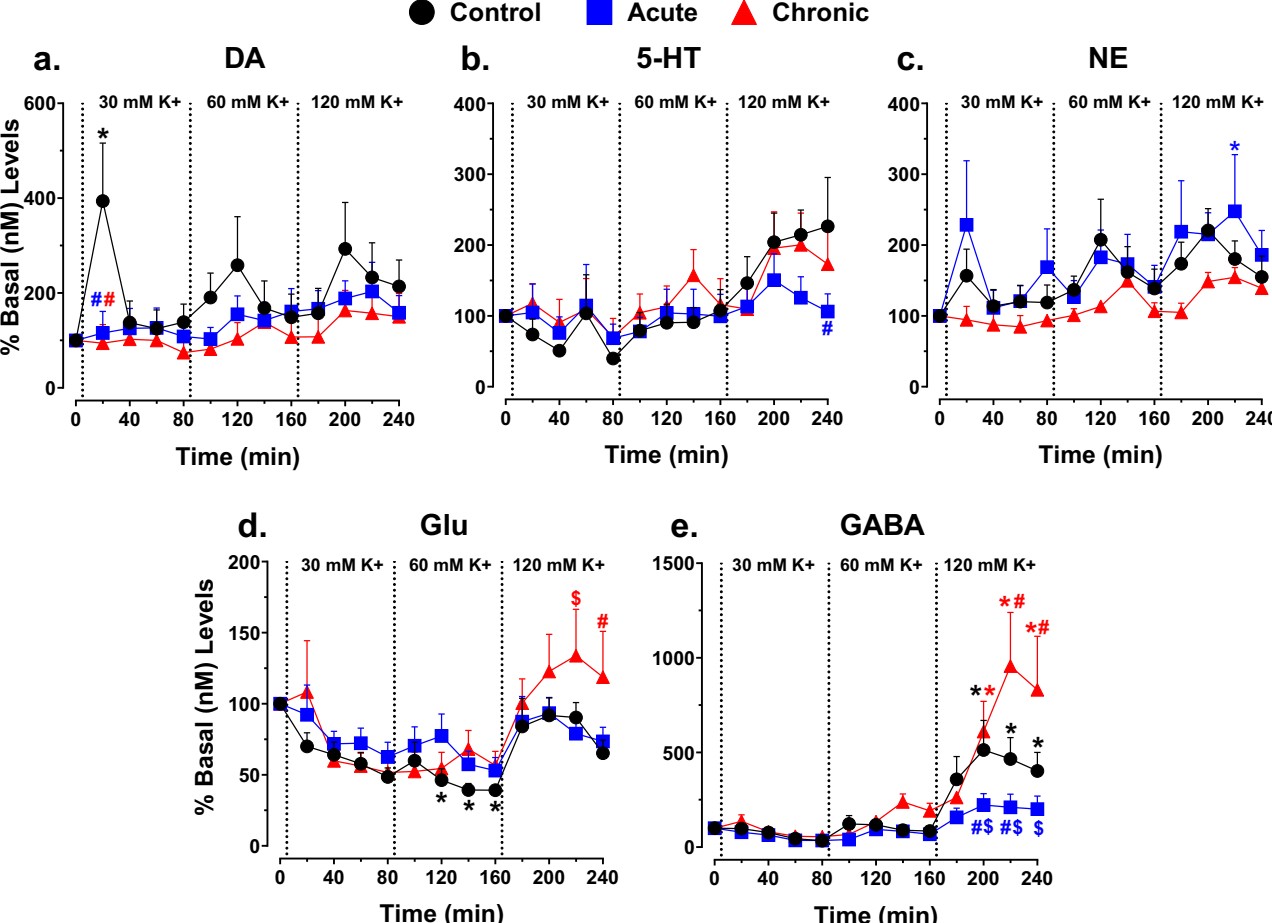

**Fig. 6 | Acute and chronic GCR exposure abolished K+-evoked DA response in PFC, but not other neurotransmitters.** Time course of the effects of K+-evoked increases in extracellular levels of DA (**a**), 5-HT (**b**), NE (**c**), Glu (**d**), or GABA (**e**) in the PFC of control ($n = 10$), acute ($n = 10$), or chronic ($n = 8$) GCR exposed mice as a % basal (nM) levels. Dialysate samples were taken from the PFC every 20 min. Each dotted line indicates time points at which K+ concentration was increased. Ordinates, percentage of basal DA, 5-HT, NE, Glu, or GABA levels; abscissae, time in minutes during K+ stimulation. Each point indicates the mean (±S.E.M.) effect shown as of percentage of basal DA, 5-HT, NE, Glu, or GABA levels; neurotransmitter levels were uncorrected for probe recovery. Time course data were analyzed using a two-way ANOVA (treatment group and concentration and treatment group and time as factors, respectively) for repeated measures over concentration and time; overall changes from basal levels determined at 4 mM K+ perfusate solution were subjected to Tukey *post hoc* analyses. Analysis of these data showed: (a) for DA: main-effect treatment, $F_{(2,325)} = 8.41$, $P = 0.0003$; main-effect time $F_{(12,325)} = 1.58$,

$P > 0.05$; a significant treatment × concentration interaction, $F_{(24,325)} = 0.81$, $P > 0.05$; (**b**) for 5-HT: non-significant main-effect treatment, $F_{(2,325)} = 1.62$, $P > 0.05$; significant main-effect time, $F_{(12,325)} = 3.93$, $P = 0.0001$; non-significant treatment × concentration interaction, $F_{(24,325)} = 0.62$, $P = 0.92$; (**c**) for NE: main-effect treatment, $F$ values$_{(2,325)} = 7.91$, $P = 0.0004$; main-effect time, $F$ values$_{(12,325)} \geq 2.40$, $P$ values $= 0.005$; non-significant treatment x concentration interaction, $F_{(24,325)} = 0.41$, $P = 0.99$; and (d-e) for Glu and GABA: main-effect treatment, $F$ values$_{(2,325)} \geq 5.06$, $P$ values $\leq 0.007$; main-effect time, $F$ values $_{(12,325)} \geq 6.23$, $P$ values $< 0.0001$; significant treatment x concentration interaction for GABA, $F_{(24,325)} = 2.74$, $P < 0001$ but not Glu, $F_{(24,325)} = 0.91$, $P > 0.05$. *$P < 0.05$ represents significant difference compared to basal values at 4 mM at time = 0. #$P < 0.05$ represents a significant difference compared to controls at each time point. $P < 0.05$ represents significant differences between acute and chronic GCR-treated mice at each time point. DA dopamine, 5-HT serotonin, NE norepinephrine, Glu glutamate, GABA γ-aminobutyric acid.

at the 120 mM K+ (Fig. 8a, b). These monoamine relationships were completely reorganized after acute GCR exposure and DA now strongly predicted 5-HT/NE levels and vice versa (Fig. 8a, b). Similarly, chronic exposure to GCR caused a strong bidirectional relationship to develop between DA and 5-HT, strengthened the 5-HT–NE interaction, and conserved the NE–DA interaction (Fig. 8a, b). In general, at the 30–120 mM, neurotransmitter networks were more balanced and well connected in control mice compared to mice exposed to acute and chronic GCR that exhibited fewer connections with more monoamine reorganization. In addition, we conducted further linear regression analysis and added a '+' sign of impact to reflect that an increase in one neurotransmitter promotes the increase in the levels of another neurotransmitter, whereas a '−' sign reflects suppression (Fig. 8a, b).

It is noteworthy that the above approaches do not treat the input data as time series. Thus, we performed a causality analysis to better

appreciate the causal impact of the neurotransmitter changes induced by different GCR exposure levels. We performed the Granger Causality test to understand the significance of those causal impacts and generated boxplots depicting the distribution of p-values over all the subjects for impact of each neurotransmitter on DA and vice versa (Fig. 9a, b). We primarily focused on DA, as our data indicate this neurotransmitter to be a key player in the reorganized neurotransmitter networks. We found that for the control group, the median p-values were greater than 0.05 for all the neurotransmitters impacting DA and vice versa (Fig. 9a, b), implying that the other neurotransmitters do not impact DA and vice versa in the control subjects. However, based on median p-values, in subjects exposed to acute GCR only the neurotransmitter GABA impacts DA (Fig. 9a) and in acute GCR group the causal impact of other neurotransmitters (except GABA) on DA cannot be meaningfully established. Moreover, Fig. 9 shows that DA in turn impacts GABA, GLU, and 5-HT, but not NE. Likewise, in

## a.

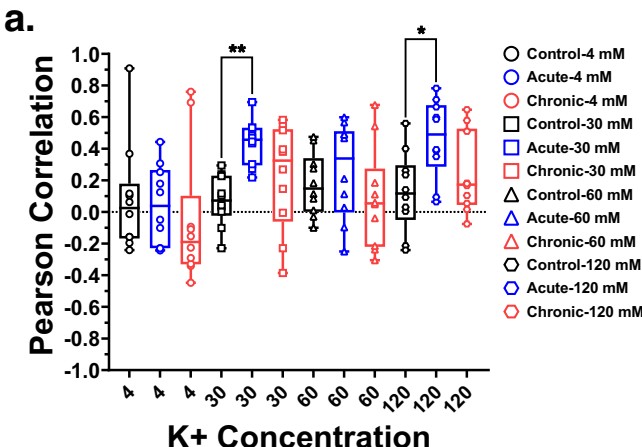

## Fig. 7 | Pearson's correlation analysis showed that acute and chronic GCR exposure differentially reorganized PFC neurotransmitter networks.

Box plot (**a**) showing overall pair-wise Pearsons correlation for each treatment group during K+ stimulation (4 mM: $F_{(2,27)} = 0.40$, $P > 0.05$; 30 mM: $F_{(2,27)} = 6.44$, $P = 0.005$; 60 mM: $F_{(2,27)} = 1.03$, $P > 0.05$; 120 mM: $F_{(2,27)} = 4.69$, $P = 0.018$). Plots show median, interquartile range, minimum, and maximum levels; One-way ANOVAs were followed by a Tukey's Multiple Comparison *post-hoc* test: *$P = 0.0153$; **$P = 0.0038$. Heatmaps (**b**) depicting strength of pair-wise neurotransmitter correlations for all the target classes and concentration levels. All analysis is based on neurotransmitter data in Figs. 5, 6: control $n = 10$, acute $n = 10$, and chronic $n = 8$. Pearson's product-moment correlation coefficient was calculated for each neurotransmitter pair to determine the relationship between each neurotransmitter. Significant change in correlation can be observed for each concentration across the target classes. Neurotransmitter networks (**c**) built using Pearson's correlation ≥ 0.5 with a significance *p*-value of <0.05. Thickness of an edge represents the strength of the correlation. Significant reorganization of networks can be observed moving across the study groups. In general, 'acute' condition shows higher connectivity among the target classes for most of the concentration levels. For all other *$P < 0.05$ indicates significant difference compared to controls. DA dopamine, 5-HT serotonin, NE norepinephrine, Glu glutamate, GABA γ-aminobutyric acid.

## b.

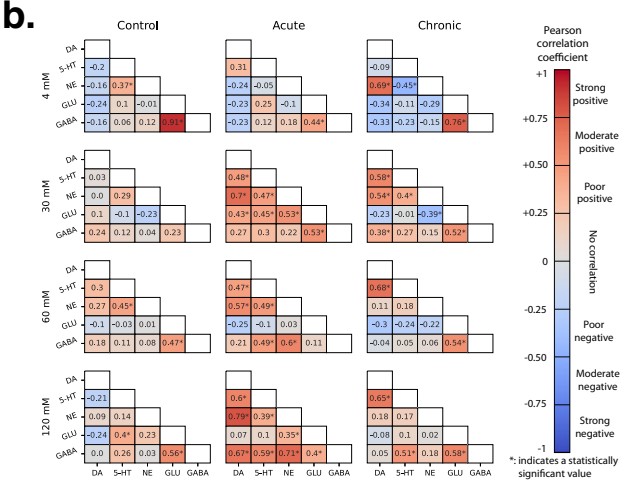

## c.

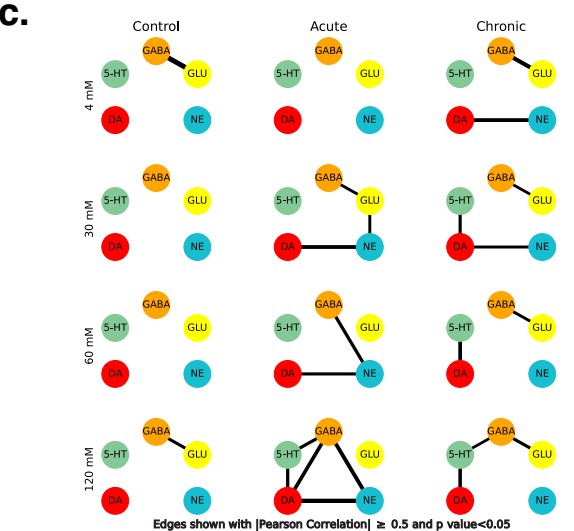

subjects exposed to chronic GCR, the median *p*-values show that GABA and NE impact DA and vice versa (Fig. 9).

## Discussion

Over the next decade, NASA's planned Artemis missions intend to establish a long-term human footprint on the Moon in preparation for future human deep space exploration to Mars. To successfully realize this goal, it is imperative to fully understand the impending CNS health risks associated with incessant exposure to deep space radiation that will be encountered by astronauts during long-duration missions to the Moon and Mars. Variations across Earth-based simulated irradiation paradigms, age, and behavioral history at the time of exposure and cognitive testing, and specifics of behavioral testing have further complicated our ability to translate findings from preclinical studies in rodents to astronauts[1–10,45]. These issues have impeded our ability to adequately document a clear link between in vivo neurobiological abnormalities and neurocognitive deficits.

Here, we evaluated whether whole-body acute or chronic exposure to a complex 33-beam terrestrial GCR simulation deteriorated performance on two distinct touchscreen tasks that appraised reward sensitivity/motivation or sustained attention in the same mature male mice irradiated at ~6 months old and, if so, whether any deficits can be linked to in vivo neurotransmitter abnormalities in the PFC. Contrary to earlier work using simplified irradiation paradigms[5–26], we found that acute or chronic GCR did not appreciably modify motivation for reward via economic demand assessments. From a risk-assessment perspective, this is a desirable outcome. Indeed, although performance on an economic demand procedure is not cognitive behavior per se, it is encouraging to document no blunted sensitivity or motivation for reinforcing consequences. Moreover, it provides evidence that other cognitive deficits, for example, the adverse effects on attentional processes and reaction time that were observed, were not simply a product of GCR-related deficits in motivation for reward. As such despite observing no deleterious effects on economic demand indices in the same mice, the development of psychomotor vigilance performance was slowed as was asymptotic reaction time levels. That is, GCR-treatment group differences subsided over time with extended daily training/testing conditions (i.e., test days 10–15). However, although differences in reaction time among treatment groups did not reach statistical criteria after day 6, in subjects exposed to acute or chronic GCR, performance on the vigilance task never fully approximated that of control animals (i.e., ~1 s inferior reaction time). Projected to astronauts, albeit statistically insignificant, these consequences could nevertheless adversely impact critical time sensitive decision making in-flight and jeopardize mission success.

The above observations are consistent with recent studies that showed exposure to simplified irradiation paradigms either selectively impaired specific aspects of cognition (attentional set shifting) or did not alter various touchscreen-based discrimination and learning assessments in male rodents[31–36]. Interestingly, Eisch and colleagues reported that single-ion exposure selectively improved touchscreen pattern separation in male mice without affecting other outcomes and improved discrimination learning in female mice but impaired

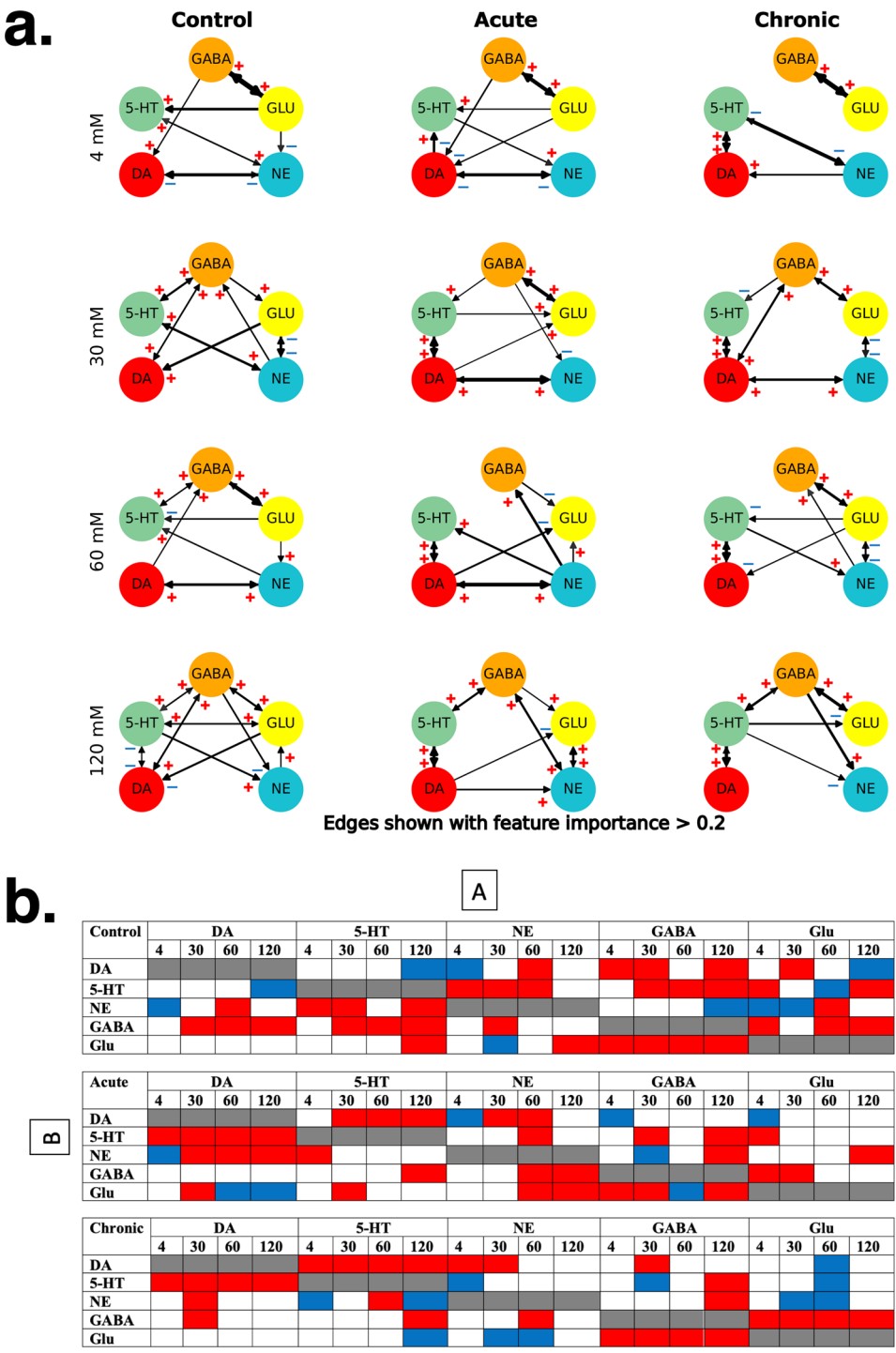

**Fig. 8 | Machine learning-derived networks unravel the impact of neurotransmitters on each other and their rearrangement after acute and chronic GCR exposure. a** Directional neurotransmitter networks built using iRF-LOOP method. The iRF network for each target class that was derived from the measured neurotransmitters as well as the absolute difference between each connection over each pair of target classes resulting in 12 directional neurotransmitter networks are shown (i.e., from 'Control','Acute','Chronic' and concentration from '4 mM','30 mM','60 mM','120 mM'). All analysis is based on neurotransmitter data in Figs. 5, 6: control $n = 10$, acute $n = 10$, and chronic $n = 8$. We also embed the derived feature importance by changing edge thickness in the networks. The thickness of the edges represents importance of one neurotransmitter level in predicting level of another neurotransmitter. Apart from the networks for the individual groups, we add panels to capture changes in connection strength between every pair of target groups (e.g., acute vs. control). A threshold of 0.2 on the feature importance values was applied so that both balanced (a neurotransmitter being equally affected by other 4) and biased connections are depicted, and faint connections are suppressed. The connection strengths above that threshold are encoded with thickness of edges. For bidirectional arrows the difference between individual directions can be understood by observing the difference in thickness of arrowheads. Significant differences in connection strength and direction can be observed moving across the three classes for each concentration level (reflected clearly by the X vs. Y panels). For example, DA-NE connection is nonexistent in 'Control' class but appears strongly in 'Acute' and 'Chronic' classes for 30 mM concentration. In addition, using linear regression analysis we have added a '+' sign of impact to reflect that an increase in one neurotransmitter promotes the increase in the levels of another neurotransmitter, whereas a '−' sign reflects suppression. **b** Illustrates an alternative representation of the neurotransmitter networks depicted through the graphs in (**a**). The boxes are marked red and blue when, respectively, a positive or negative A to B connection exists with an importance greater than 0.2. DA dopamine, 5-HT serotonin, NE norepinephrine, Glu glutamate, GABA γ-aminobutyric acid.

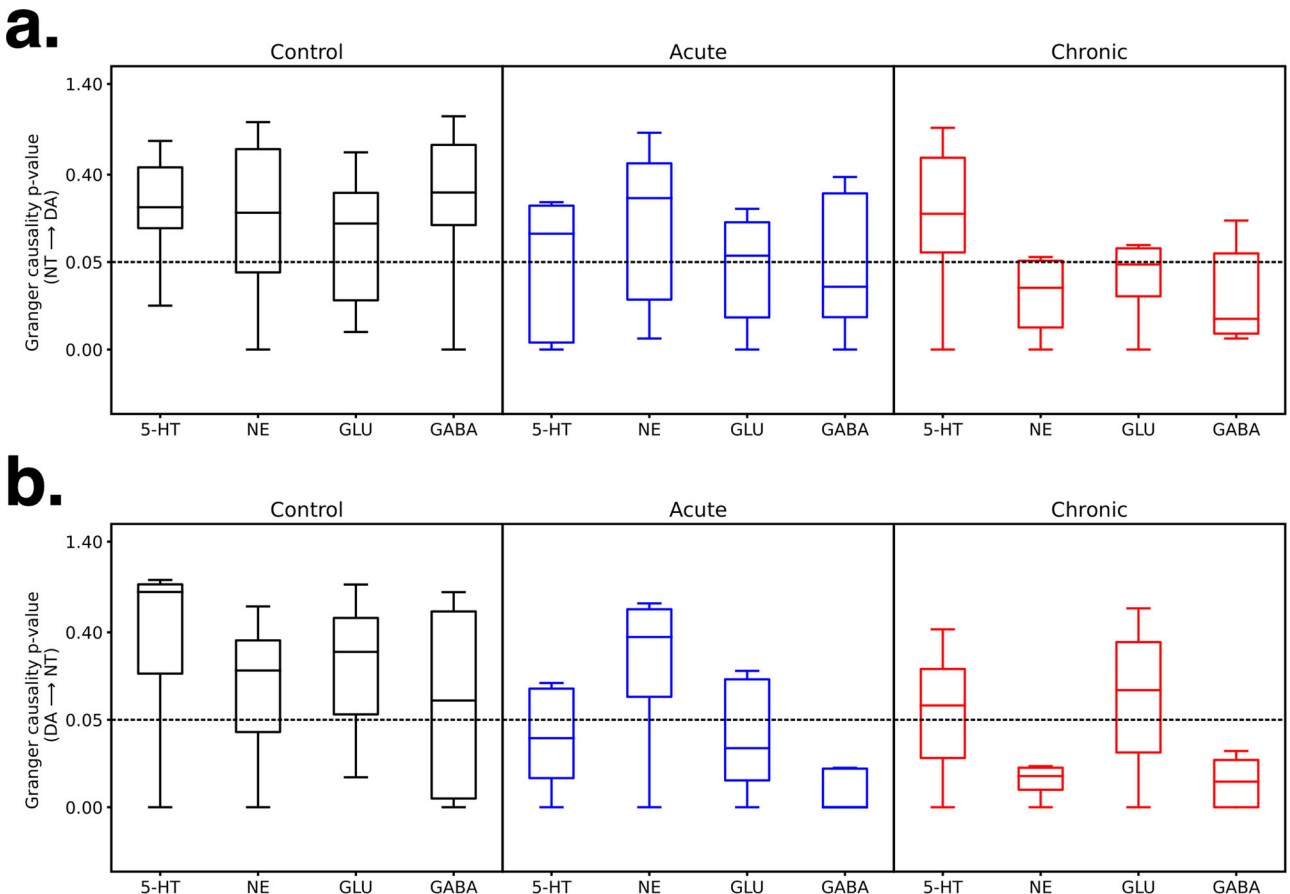

**Fig. 9 | Pair-wise Granger causality tests identify dopamine as an important player in driving or being driven by other neurotransmitters after acute and chronic GCR exposure. a** The Granger Causality test to understand the significance of causal impacts with GCR exposure vs controls and generated boxplots depicting the distribution of *p*-values over all the subjects for impact of each neurotransmitter on DA. **b** The Granger Causality test to understand the significance of causal impacts with GCR exposure vs controls and generated boxplots depicting the distribution of *p*-values over all the subjects for DA's impact on each neurotransmitter. All analysis is based on neurotransmitter data in Figs. 5, 6: control $n = 10$, acute $n = 10$, or chronic $n = 8$. The boxplots depict the median (middle line), 25th percentile (Q1; lower boundary of the box), 75th percentile (Q3; upper boundary of the box), and lowest datum above Q1−1.5*(Q3−Q1) and highest datum below Q3 + 1.5*(Q3−Q1) as whiskers. Outliers are not shown. DA dopamine, 5-HT serotonin, NE norepinephrine, Glu glutamate, GABA γ-aminobutyric acid.

stimulus-response rule-based habit learning[35,36], suggesting certain sex differences. Work published from a similarly exposed cohort found female mice somewhat more resistant to cognitive deficits[34], which paralleled past results obtained with single helium ion exposures[46]. This work has suggested that the higher levels of baseline inflammation may precondition the female rodent brain, thereby attenuating the extent of space radiation injury and possibly ameliorating resultant cognitive impairments.

In previous work, cohorts of male and female mice exposed to the same acute or chronic delivery of the 33-beam GCR were found to exhibit similar and less severe neurocognitive and electrophysiological impairments[34]. Importantly, this study was not designed to discriminate differences in dose rate but rather to elucidate how a complex GCR simulation might impact CNS functionality. For the chronic irradiations, dose rates for individual energetic particles varied between ~0.04 and 0.9 cGy/min, with a collective mean dose rate of ~2.08 cGy/h. For the acute irradiations, dose rates for individual energetic particles varied between ~0.16 and 4 cGy/min, with a collective mean dose rate of 25 cGy/h. Unfortunately, behavioral tasks are simply not sensitive enough or designed to pick up these small differences in radiation dose rates. It is important to note that few studies have systematically and rigorously evaluated the impact of space relevant dose rates on CNS functionality. The first study that implemented a dose rate of 1 mGy/day over the course of six months, reported an array of adverse behavioral,

electrophysiological, and molecular outcomes compared to unirradiated controls[47].

Collectively, these results suggest that exposure to space-relevant radiation might be less detrimental on neurocognitive functionality than expected from earlier reports using more simplified irradiation paradigms[34]. The complexity and differences in CNS outcomes across studies highlights the critical need for NASA to further evaluate the impact of space-relevant radiation exposures on multiple within-subject neurocognitive endpoints in male and female subjects. Adapting approaches used here can only improve estimates of space radiation exposure risk to the neurological well-being of astronauts[9,31–33,35,36,45,48].

Several aspects of the present results are noteworthy. First, the finding that vigilance performance improved after acute or chronic GCR exposure with extended daily touchscreen training/testing is intriguing. Here, mice were first trained/tested on the economic demand task followed by the vigilance task. Projecting these data to astronauts suggests that tasks involving touchscreen-based vigilance training or even gaming might help forestall further cognitive decline caused by GCR. In support of the foregoing, data from human studies show that virtual environmental enrichment can enhance hippocampal memory across astronaut relevant age groups[49,50]. Second, it is possible that the detrimental effects of both acute or chronic GCR on both touchscreen tasks may have dissipated and might be more pronounced early (i.e., 1–2 months) after GCR exposure. Although, additional work is needed to document these early effects of acute or

chronic GCR exposure, the adverse effects on attention observed 5−6 months post-GCR exposure are consistent with prior work showing that harmful CNS effects persist long (i.e., up to 1 year) after irradiation[7–12,21–24]. Thus, the adverse cognitive effects of GCR may persist during long-duration missions and upon return to Earth, which might compromise subsequent terrestrial activities. Third, the same mice were tested on two distinct tasks that varied in the type of cognitive challenges required to overcome for optimal performance, i.e., obtaining reward for increased price vs maintaining attention to obtain reward. Prior studies have shown that adjusting conditions that varied in cognitive difficulty could reveal more subtle yet significant effects of radiation exposure on cognition that might not otherwise be observed using simple tasks[6–9,45,48]. Future studies that manipulate cognitive load in subjects pre-trained on neurocognitive tasks will be helpful for elucidating spaceflight stressor effects on astronaut performance during (i.e., evolving mission scenarios) and after the mission[6–9,45,48].

Although past studies have related neurocognitive deficits to compromised structural plasticity, network level connectivity, elevated microgliosis, astrogliosis, and/or inflammatory signaling in the irradiated brain[5–26], the precise neurobiological mechanisms responsible for GCR-induced neurocognitive damage remains uncertain. Given that brain neurotransmitters provide relevant readouts of neurocognitive function[27], it is surprising that the impact of space-relevant radiation on brain neurotransmitter dynamics has not been delineated. Here, we focused on identifying a link between GCR-induced changes in reward sensitivity and attention and in vivo neurotransmitter alterations in the PFC, a key brain region involved in both reward and attention[27,37–41]. We found that exposure to acute or chronic GCR selectively increased normal basal levels of DA, 5-HT, and NE, but not Glu or GABA compared to controls. These results are consistent with previous in vitro findings in rodents showing long-lasting alterations in DA, other monoamines, and their metabolites after exposure to simplified irradiation paradigms, but are somewhat contrary to alterations in Glu and GABA activity observed following simplified radiation exposures[8,14,15,17–20,33]. From a broader perspective, previous studies have used simplified single-ion exposures which may not produce the physiologically relevant changes desired to simulate deep space radiation. By using a complex 33-beam GCR simulation we more accurately reproduce the physiologically relevant conditions, which may produce differing results from simpler, less representative models. However, we emphasize that past work using complex GCR[34], mixed field 5 ion studies[26,51] and even acute[52] and chronic low dose neutron[47] studies suggest multiple disruptions across interconnected brain regions. Moreover, chronic GCR exposure increased levels of all K+-stimulated neurotransmitters compared to controls, indicating global PFC changes that compromised all neurotransmitter systems; these effects were less significant after acute GCR. Notably, both GCR paradigms similarly altered DA levels compared to other neurotransmitters. In addition, time course analysis showed that the ability of the PFC DA system to respond normally to changing physiological conditions was severely and equally compromised after exposure to both paradigms, where mice were unresponsive to K+ stimulation. In contrast, the temporal patterns of other neurotransmitters were less affected, requiring greater K+-stimulation to observe any differences. Thus, we posit that acute and chronic GCR did not alter reward sensitivity/motivation because significant amounts of PFC DA was available to engender rewarding properties. However, the adverse effects on attention might be attributed to radiation-induced changes in the temporal pattern of DA dynamics in the PFC, i.e., compromised normal DA responsiveness that is typically necessary for attentional processes. This is consistent with the view that optimal extracellular DA levels in the PFC is critical for sustaining normal neurocognitive functioning, and that any DA disruptions can lead to cortical dysfunction[42,43]. That DA mechanisms are compromised to differentially impact aspects of neurocognitive function is in agreement with prior work showing

associations between changes in DA activity and some but not all DA-mediated behaviors[8,13–15].

It is important to note that the other factors (e.g., behavioral training and age) in addition to GCR exposure could also contribute to the overall observed PFC neurotransmitter changes. While assessment of age and behavioral training history were not part of the overall study goals, it is noteworthy that the animals in the control group were of the same age with the comparable behavioral history as the GCR-exposed animals, meaning that while age and behavioral training may be important variables, it is one that we considered in the experimental design. Nevertheless, we appreciate that the neurotransmitter responses may be a consequence of how these controlled variables may interact with the neurotransmitter changes elicited by GCR simulation. From a more global perspective, the entire brain is susceptible to radiation injury, but clearly will not exhibit equivalent levels of sensitivity. Behavior is a multifaceted response to a variety of molecular and biochemical changes impacting neurotransmission, and GCR exposure produces persistent changes in functional and molecular outcomes in the CNS. Other work has found exposure to energetic iron ions to cause persistent reductions in the glutamatergic readily releasable pool in rat hippocampal synaptosomes[18]. Here complex GCR exposures may portend changes in neurotransmitter availability and/or alter the dynamic response of neurotransmitters available which can greatly impact neurocognitive function. Increased levels of neurotransmitters (Fig. 5) may well predispose certain excitatory and inhibitory circuits to fire asynchronously, which may not lead to overt changes in behavior under basal operating conditions, but under situations of exaggerated spaceflight stress, critical, accurate and timely decision making may be compromised. For example, if we translate increasing physiological stress (i.e., K+ levels) to increasing levels of spaceflight stress, then muted dynamic neurotransmitter responses (particularly with DA, Fig. 6a) could easily compromise reaction times and adaptive decision making, a central role for the PFC. Together these data suggest that interventions targeting DA pathways may help forestall and/or restore homeostatic signaling in the irradiated brain[30]. These are the first within-subject in vivo studies in mice to document the impact of space-relevant GCR exposures on multiple touchscreen-based neurocognitive assays and associated neurotransmitter dynamics in the PFC.

The PFC consists of a complex network of multiple neurotransmitter systems that modulate each other's actions and are interconnected across brain regions to maintain normal CNS function, including reward/motivation and attention[27,42,43]. Indeed, aberrant neurotransmission in the PFC has been closely linked to various neurological diseases[27,42,43] raising concerns that neurotransmitter disruptions may adversely impact astronaut health/well-being during and after deep space missions. Since it is likely that our measurements reflect global changes in neurotransmitter availability and neurocircuit activity, we embarked on a detailed analyses of pairwise neurotransmitter networks. Pearson's correlation analysis revealed a profound reorganization of relationships between PFC neurotransmitters following acute or chronic GCR exposure relative to controls. In particular, the pairwise relationships at each K+-evoked concentrations were reorganized and strengthened among the monoamines DA, 5-HT, and NE as well as monoamines and GABA, with some weakening in the pairwise relationships between monoamines and Glu following both GCR paradigms compared to controls (Fig. 7). Collectively, our findings of disruption in monoamine neurotransmitter networks from correlation analyses were corroborated by our predictive machine learning model that uncovered significant differences in PFC monoamine neurotransmitter networks among groups both in terms of connection strength and causation (Fig. 8). More specifically, a pronounced increase in the impact of 5-HT or NE on DA and vice versa coupled with a significant weakening of GABA and Glu's impact on these monoamines in the irradiated PFC compared to controls. This is

in line with the existence of functional interactions between 5-HT and DA where 5-HT either positively or negatively modulates DA levels to impact cognition[44]. The capability of GCR exposures to elicit marked neurotransmitter network reorganization is noteworthy and certainly not restricted to the PFC, and likely represents some of the fundamental biochemical changes that are contributory if not causal to space radiation-induced behavioral and neurocognitive decrements.

The Granger Causality test provided deeper insight into the PFC neurotransmitter network that was altered over protracted acute and chronic post-exposure times. Evident from this analysis is that DA has a significant impact on irradiated neurotransmitter levels (most notably GABA, Fig. 9a), while the other neurotransmitters also impact DA levels (most notably after chronic exposures, Fig. 9b) when compared to controls. The reciprocity of the neurotransmitter network strengthens our arguments for the functional reorganization of these networks following space radiation exposure. While it remains difficult to pinpoint the incipient event that triggered such widespread change, it may also prove relatively unimportant in the context of chronic exposures encountered during extended deep space travel. We are also cognizant that changing levels of K+ might have off-target or more subtle effects on network connectivity that may confound current interpretations, especially at high levels of K+ stimulations that may have induced oscillations in the GLU-GABA networks. To the extent that our machine learning and causality analyses are predictive of network-level reorganization, then it seems plausible that such long-term alterations will be an inevitable consequence of long-duration space travel. What is less certain, is whether such reorganization of neurotransmitter interactions will continue to evolve and if/how they will functionally impact critical decision making and adaptive reasoning under more stressful situations in space.

It is possible that probing neurotransmitter network changes in the PFC at multiple time points (e.g., early, middle, and late) after GCR exposure may reveal further insights into how in vivo neurotransmitter network dynamics are altered over time to impact behavior. Unfortunately, such an endeavor would require a substantially larger cohort of subjects that were simply not available and therefore was beyond the scope of this work. Nevertheless, it is noteworthy, that our data showing long-lasting impact of acute and chronic GCR exposure on behavior and neurotransmitter consequences is highly consistent with the long-term adverse effects on other behavioral and molecular endpoints that have been documented following exposure to space radiation[7–12,21–24]. Thus, the data presented here sets a solid benchmark for an expanded evaluation of in vivo neurotransmitter network dynamics over time following GCR exposure. From a broader perspective, it is likely that the neurotransmitter network reorganization may be different immediately after GCR exposure vs at later time points. The detrimental effects of both acute or chronic GCR on both touchscreen task and neurotransmitter function may have dissipated and might be even more pronounced early after GCR exposure. However, in the context of a long duration spaceflight, the changes we measure are likely more reflective of the accumulated GCR dose and acute changes (if they do occur) have not been documented and would not translate to late onset radiation-induced cognitive decrements known to take time (months) to manifest after such low dose complex radiation exposures. We therefore contend that our protracted measurements are much more relevant to NASA risk assessments where potential decrements are more likely to manifest on the return from Mars rather than en route to Mars. Thus, while late changes in neurotransmitter levels may reflect the combined impact of all experimental stressors, under our experimental conditions, we believe these are most likely the result of permanent radiation-induced changes to the basal neurochemical state of the irradiated brain.

The fact that low dose, space relevant GCR exposures can elicit functional reorganization of neurotransmitter networks in a single brain region is noteworthy and suggests that similar changes likely transpire throughout the brain after whole body exposures. Indeed, smaller scale changes in pair-wise cell recordings conducted in the hippocampal CA1[17,26,53] have not revealed the more global network level disruptions found after larger scale recordings across multiple brain regions[23] or by LTP mesurements[34,47,51]. Importantly, at the space relevant doses used (≤0.5 Gy), cell death in not the cause of the functional changes reported here, or in past studies conducted under similar dosing paradigms[1–8]. This raises the intriguing idea that the whole brain may be considered the critical target for understanding general CNS radiation effects. Under this conceptual framework, much of current and past studies can be placed in context, where global disruptions in neurotransmitter networks alter synaptic connections across broad brain regions to elicit changes in functional outcomes that cannot be rationalized by smaller scale measurements. Thus, these findings of how the PFC neurotransmitter networks are reorganized by realistic complex GCR simulations provide insights into how such changes impact diverse neurocognitive outcomes and could help re-shape how we envision the impact of space radiation on CNS functionality.

## Methods

### Subjects and irradiation

All animal procedures were approved by the Institutional Animal Care and Use Committee at McLean Hospital and NASA Space Radiation Laboratory (NSRL) at Brookhaven National Laboratory and were conducted in accordance with guidelines provided by the Committee on Care and Use of Laboratory Animals of the Institute of Laboratory Animal Resources, Commission on Life Sciences, National Research Council[54].

Experimentally-naïve male C57BL/6 mice (Jackson Laboratories, Bar Harbor, MA) ~6–9 months old (~30–40 years in human age) and weighing 26–35 g at the time of testing were used. Mice were exposed to a complex 33-beam composition of galactic cosmic radiation simulation (GCRSim) in the NASA Space Radiation Laboratory (NSRL) at Brookhaven National Laboratory (BNL, Upton, NY). The ionic composition of GCRSim contained a mixture of ion species consisting of protons, helium, oxygen, silicon, titanium, carbon, and iron that recapitulated many of the most abundant elements in GCR[7]. Separate groups of mice ($n = 8–10$/group) were exposed to either an acute or chronic GCRSim irradiation regimen as described previously[34]. Subjects in the acute group were exposed to the 33-beam GCRSim delivered in a single day over a 2-hr period for a total dose of 40 cGy. Mice in the chronic group were exposed to the same 33-beam GCRSim delivered over ~1 h for 4 weeks (6 days/week) for a total dose of 49.9 cGy (2.08 cGy/day). The total dose delivered is similar to what an astronaut would experience during a voyage to Mars[1–8]. Individual mice were loosely restrained within acrylic enclosures (3 × 1.5 × 1.5 in) mounted perpendicular to the beam line during the duration of whole-body exposures. Irradiation was overseen by NSRL staff, who also performed all radiation dosimetry and confirmed spatial beam uniformity. The low total GCR doses administered produced no observed changes in body weight. Further details on the operation of the NSRL facility have previously been described[55]. Age-matched control mice underwent all aspects of the study in parallel to those receiving irradiation, were housed under similar conditions and handled equivalently, aside from not receiving GCR irradiation. Following completion of irradiation procedures, all mice were transported from NSRL BNL to McLean Hospital (Belmont, MA) and underwent an 8-week quarantine period upon arrival. Thereafter, mice were housed in a temperature-(~18–23 °C) and humidity-(~40–60%)-controlled vivarium with a 12-h light/dark cycle (lights on from 7 AM to 7 PM). Subjects were each fed 2.5–3.0 g of rodent chow post-session and had unrestricted access to water in their home cage. Experimental sessions were conducted five days a week (Mon-Fri). All behavioral and neurochemical testing in mice was conducted over the course of several months

(i.e., 6–9 months old) post-irradiation and post-quarantine period at McLean Hospital.

## Behavioral procedures

**Apparatus.** Details and schematics of the touch-sensitive experimental chamber for rodents can be found in Kangas and Bergman[56]. Briefly, a custom-built Plexiglas chamber ($35 \times 30 \times 25$ cm) was situated in a sound- and light-attenuating enclosure ($60 \times 40 \times 45$ cm). A 17" touch-sensitive screen (1739L, ELO TouchSystems, Menlo Park, CA) was mounted on the right-hand inside wall of the chamber. An infusion pump (PHM-100-5, Med Associates, St. Albans, VT) outside the enclosure was used to deliver a sweetened condensed milk solution into the shallow reservoir (diameter: 3 cm) of an aluminum receptacle (5x4x1 cm) that was mounted slightly above the Plexiglas floor and centered on the left-hand inside wall. A speaker bar (NQ576AT, Hewlett-Packard, Palo Alto, CA) mounted above the touchscreen was used to emit audible feedback. White noise was present outside the chamber to mask extraneous sounds. All experimental events and data collection were programmed in E-Prime Professional 2.0 (Psychology Software Tools, Inc., Sharpsburg, PA).

**Touchscreen training.** Modified response shaping techniques were first used to train mice to engage with the touchscreen[57]. Subjects first learned to touch a $5 \times 5$ cm blue square on a black background centered on the touchscreen, with its lower edge 1.5 cm above the floor. This required the mouse to rear on its hind legs and make a response on the box with its paw. Each response was followed with 0.02 ml of a 20% sweetened condensed milk concentration, paired with a 600-ms yellow screen flash and a 440 Hz tone, and followed by a 5-s blackout period. Following reliable responding, presentation of the box alternated in a quasi-random fashion across one of three positions: center, 3.5 cm to the left of center, or 3.5 cm to the right of center. After subjects completed two consecutive 100-trial training sessions within 60 min, the economic demand task commenced.

**Economic demand task.** Consumption of varying milk concentrations was assessed using a within-session economic demand task to determine the effects of GCR exposure on sensitivity to reward. Demand for 4 different milk concentrations were each tested during 3-session conditions in all mice using the following fixed sequence: 20%, 5%, 50%, and 0% sweetened condensed milk in water. Reward delivery was contingent on responses to a centered $7 \times 7$ cm green box on a purple background and consisted of 0.02 ml milk paired with a 600-ms yellow screen flash and a 440 Hz tone, followed by a 10-s blackout period. The response requirement increased in an ascending order (2, 5, 10, 30, 50) within-session across five 10-min components that were separated by a 2-min inter-component blackout periods.

**Psychomotor vigilance task.** After completion of the economic demand task, subjects were exposed to a touchscreen-based variant of a psychomotor vigilance task to evaluate the effects of GCR exposure on attentional processes. This task variant is based on previous studies examining attentional processes by responding to blips on a radar screen that vary in location and intermittency[58]. Trials began with the presentation of a $7 \times 7$ cm stimulus (pink square) on the screen (black background) in one of six locations (evenly spaced from left to right, 2.5 cm above the floor). The intermittency of stimulus presentation (after a 15-, 30-, or 45-s intertrial interval) as well as the stimulus location were randomized across trials. The duration of stimulus presentation on the first trial of the first session was 10 s for all subjects. Touching the stimulus within this set duration resulted in delivery of 0.02 ml of 20% milk (standard concentration for mice) paired with a 600-ms yellow screen flash and 440 Hz tone. The maximum stimulus duration on the following trial then decreased by 0.25 s. If the subject failed to respond to the stimulus before the duration had elapsed, the

stimulus disappeared, and the maximum stimulus duration increased by 0.25 s on the following trial. Daily sessions were comprised of 48 trials and the titrated duration of stimulus presentation on the last trial of a session determined the initial stimulus duration for the next session. There was no upper limit for the stimulus duration, but the lower limit was set to 0.25 s to avoid a 0-s duration stimulus presentation. These titrating contingencies were designed to dial in the attentional capabilities of the subject via their reaction time and ability to maintain task performance across a 48-trial session (i.e., focused vigilance) by making the durations of stimulus presentations briefer as the subject mastered the task. This also allowed for the assessment of potential GCR-treatment effects on both task development as well as asymptotic attentional performance. Each subject completed a total of 15 sessions under these task conditions.

## Data analysis

**Economic demand.** Individual subject and group data from the final session of each condition were fit according to the exponential model of demand as described by Hursh and Silberberg[59]:

$$\log Q = \log Q_0 + k(e^{-aQ_0 C} - 1) \qquad (1)$$

where $Q$ is quantity consumed, $Q_0$ is consumption as price approaches 0 (demand intensity), $k$ is a constant representing the range of consumption in log units, $\alpha$ is the rate of change (elasticity) in consumption as price goes up, and $C$ is cost (response requirement). To account for cases of zero consumption that make logarithmic transforms impossible, 0.1 was added to all consumption values. Group demand functions were normalized across milk concentrations to account for differences in individual rates of responding using previously described methods[60]. Consumption is thus expressed as a percentage of $Q_0$ and price is expressed as the response requirement to produce 1% of $Q_0$ ([response requirement*$Q_O$]/100).

Nonlinear regression using Eq. 1 with nonnormalized values derive demand parameters that quantitatively define different aspects of reinforcing efficacy, such as demand intensity ($Q_0$), essential value ($1/\alpha$), and goodness of fit ($R^2$)[61]. The standard error for $Q_0$ and essential value parameters (presented as error bars in Fig. 2c) were computed in GraphPad Prism 9 (La Jolla, CA) using the following equation: SE(Pi)= sqrt[(SS/DF)*Cov(i,i)], where Pi is $i$th adjustable (non-constant) parameter, SS is sum of squared residuals, DF is degrees of freedom (the number of data points minus number of parameters fit by regression), Cov(i,i) is $i$th diagonal element of covariance matrix, and sqrt() is square root. To evaluate the statistical significance of GCR exposure on economic demand parameters, data were subjected to two-way analysis of variance (ANOVA).

**Psychomotor vigilance.** For the psychomotor vigilance task, mean titrated reaction times across each 48-trial session served as the primary dependent measure to capture the subject's vigilant performance during this task. Titrated reaction times is a metric designed to capture session-wide performance on this task and is comprised of averaging the reaction times to respond to the stimulus during trials where they successfully attended (i.e., hits) and, to factor in failed attempts to attend and respond in time (i.e., misses), the full duration of stimulus presentation on those trials. These values were obtained for each subject and contributed to the group mean (±SEM), which were compared statistically via two-way ANOVAs followed by a post-hoc Tukey test when appropriate. It is important to note that unlike several common psychomotor vigilance task variants where premature responses are a critical metric, this was not a dependent measure in the present studies. This is because the touchscreen-based task variant does not have fixed operanda (i.e., nose-poke holes or levers) to respond prematurely on, but rather stimuli appear and disappear on randomized screen locations. As such, premature responses

were not possible and responses to the blank touchscreen were not recorded.

## Neurochemical procedures

After completion of the economic demand and psychomotor vigilance task, studies were conducted in all subjects to determine how GCR exposure impacts neurochemical signatures in the PFC.

**In vivo microdialysis in freely moving mice.** The general surgical, microdialysis probe, experimental, and histological procedures have been described previously (see Fig. 4 for experimental design and set-up[62,63]. Briefly, under an anesthetic mixture of ketamine and xylazine (120 and 16 mg/kg, respectively, administered i.p.), C57BL/6J male mice were implanted with concentric dialysis probes, aimed at the PFC based on previous studies and according to the mouse brain atlas by Paxinos and Franklin[64] (uncorrected coordinates Anterior $(A) = +$ 2.2 mm, Lateral $(L) = \pm 0.3$ mm, Vertical $(V) = 2.5$ mm; anterior and lateral coordinates were measured from bregma, and vertical coordinate was measured from the dura). Concentric dialysis probes were assembled with AN69 dialyzing membranes with a 10–20 kDa molecular cutoff (Hospal Dasco, Lyon, France) as described previously[62,63]. Note that the molecular weight cut-off for the detected neurotransmitters in this manuscript as well as typical neurotransmitters and their metabolites is usually less than 1 kDa. Two pieces of silica-fused capillary tubes serving as an inlet (set at 6 mm from needle tip) and outlet (set at 5 mm from needle tip) tubing of the probe were inserted into a 22-gauge stainless steel needle and fixed into place by glue. The inlet and outlet tubing were inserted into a 4-mm capillary (0.25-mm external diameter) AN69 dialyzing membrane that was enclosed with a drop of glue; the inlet tubing was set at approximately 0.1 mm from the closed end of the dialyzing fiber. Next, the dialyzing membrane was fixed to the inlet and outlet tubing with glue that limited the exposed dialyzing surface of membrane to the lower 1-mm of probe, i.e., the space between the inlet and outlet not covered by glue. All dialysis probe materials were glued to a custom-designed 3D-printed dialysis probe casing.

After surgery, mice were placed into hemispherical CMA-120 cages (CMA/Microdialysis AB) and allowed to recover overnight. The CMA-120 cages were equipped with overhead fluid swivels (Instech Laboratories Inc., Plymouth, PA) for connection to the dialysis probes. Approximately 20 hr after implantation of probes, microdialysis studies were conducted on freely moving mice. Ringer's solution (147.0 mM NaCl, 2.2 mM CaCl$_2$, and 4.0 mM KCl) was delivered at a constant flow rate of 2.5 μL/min through the dialysis probes using a 2.5-mL syringe (Hamilton Co., Reno, NV) attached to a CMA/102 Microdialysis Pump (CMA Microdialysis AB). The first dialysate sample (50 μL) was taken approximately 30 min after the pump was started, and thereafter samples were taken every 20 min. All samples were immediately frozen, then stored at −80 °C until analysis (see below). A total of 4 baseline samples were collected. The ringer solution consisting of 4.0 mM KCl is typically considered to be comparable to artificial cerebrospinal fluid in microdialysis studies and has been used, with some minor variations (2.7–4.8 mM K+), by us and others for decades as the standard perfusate for establishing in vivo baseline levels of neurotransmitters in studies assessing effects of various stressors (environmental, age, pharmacological)[62,63,65–71]. Moreover, the baseline values observed at 4 mM K+ include the effect of GCR stimulation and do not attempt to establish a generalizable baseline for all animals, rather they establish what GCR vs control groups neurotransmitter levels are as baseline due to the differing conditions.

Immediately after completion of baseline sample experiments, the effects of high K+-induced changes in neurochemical levels were tested by substituting the Ringer's solution with a perfusate consisting of the following three different ion concentrations: 30 mM KCl, 121 mM NaCl, and 2.2 mM CaCl$_2$ delivered for 80 min followed immediately by

60 mM KCl, 91 mM NaCl, and 2.2 mM CaCl$_2$ delivered for another 80 min, followed by 120 mM KCl, 31 mM NaCl, and 2.2 mM CaCl$_2$ delivered for a last 80 min time period. The low to high K+ concentrations were selected not to reproduce functionally relevant physiological conditions, but to perturb the system in a manner that may elucidate more subtle changes that would not otherwise be revealed at lower levels of K+-stimulation as well as more obvious alterations in the neurotransmitter network dynamics that are apparent at high K+ stimulation. This powerful approach permits the examination of a K+-evoked concentration dependent (dose–response curve) change in neurotransmitter responsiveness and allows us to document differing neurotransmitter dynamics at different levels of K+ stimulation. Note that this approach of applying low to high K+ stimulation (100–120 mM) through the microdialysis probe is commonly used within the literature to assess changes in brain neurochemistry[65–74].

Following completion of the experiment, mice were euthanized, brains were removed and fixed in 4% formaldehyde solution. Fixed brains were then sliced by vibrating microtome (Microm HM 650 V; Microm International, Thermo Scientific) into 30–50 μm serial coronal sections, oriented according to the mouse brain atlas[64]. Probe location was identified via microscopic examination of coronal sections, and the dialyzing portion of all probes implanted in the PFC as shown in Fig. 4b, c. For simplicity, all probe placements are shown on only one side, whereas the hemisphere into which the probe was implanted varied between left and right sides randomly across subjects.

**Analytical procedure.** LC-MS grade acetonitrile (ACN), formic acid (FA) and HPLC grade water were purchased from Thermo Fisher scientific (NJ, USA). Benzoyl chloride and $^{13}C_6$-benzoyl chloride was purchased from Sigma Aldrich (St. Louis, MO). All analytes and neurotransmitters were purchased from Sigma Aldrich unless indicated otherwise. Stock solutions of, dopamine (DA), γ-aminobutyric acid (GABA), glutamate (Glu), 5-hydroxytryptamine (5-HT), and norepinephrine (NE) (Cayman Chemicals, Ann Arbor, MI), were made in HPLC grade water and kept at −80 °C. A standard mixture was diluted from stocks with HPLC grade water, aliquoted, evaporated and stored at −80 °C until analysis. Single aliquots of calibration standard solution were reconstituted in ringer's solution, (147.0 mM NaCl, 2.2 mM CaCl$_2$, and 4.0 mM KCl) adjusted to pH 7.4 with sodium hydroxide (NaOH). Derivatized internal standard stocks were frozen at −80 °C and reconstituted in ringer solution to prepare working stock, then diluted in a 1:4 ACN: water solution containing 2% (v/v) sulfuric acid and used as final quench solution in derivatization reaction. Benzoyl chloride solutions were made fresh for each analysis.

Major neurotransmitters were quantified in mice dialysate samples by Liquid Chromatography Mass Spectrometry (LC-MS/MS) following benzoyl chloride derivatization procedure as described previously by Kennedy and co-workers[75,76]. Immediately prior to analysis, samples were thawed at room temperature. 40 μL of sodium tetraborate buffer (pH 9.5) and 20 μL of benzoyl chloride (2% solution in ACN) were added sequentially to 20 μL of thawed dialysate sample. The solution was vortexed, and the benzoylation reaction was allowed to proceed for 10 min at room temperature. Finally, the derivatization reaction was quenched with 20 μL of internal standard solution, prepared as described below. Following vortexing, the derivatized solution was transferred to sample vials and subjected to LC-MS/MS analysis.

The internal standard solution was prepared by the same derivatization procedure as described above, using $^{13}C$ labeled benzoyl chloride (2% solution in ACN) and a standard solution of analytes, with individual analytes falling between 8 and 80 μM. Derivatized internal standard mixture was further diluted in 20% ACN containing 2% sulfuric acid before addition as a quenching solution to sample

reactions. All samples were analyzed on Thermo Fisher TSQ Altis Triple Quadrupole mass spectrometer with a Thermo Fisher Vanquish UHPLC at the front end. The derivatized neurotransmitters were first separated on Phenomenex C18 Kinetex F5 column (150 ×4.6 mm, 2.6 μm) and then analyzed by mass spectrometer operated in ESI-positive multiple reaction monitoring mode.

**Data analysis.** Data from in vivo microdialysis experiments are expressed as cumulative increases in extracellular levels (nM) for each neurotransmitter at each K+ perfusate solution. In addition, time course data for each neurotransmitter are expressed as a percent of basal (4 mM K+) levels of each neurotransmitter. Basal levels were calculated as the mean of values from all four consecutive samples that were taken during stimulation with 4 mM K+. All results are presented as group means (±S.E.M.). An unpaired two-tailed $t$ test was used to determine differences in cumulative increases in neurotransmitter levels among groups. Time course data were analyzed using a two-way ANOVA (treatment group and concentration and treatment group and time as factors, respectively) for repeated measures over concentration and time; overall changes from basal levels determined at 4 mM K+ perfusate solution were subjected to Tukey *post hoc* analyses (GraphPad Prism, version 9.1.3). The Pearson's product-moment correlation coefficient was calculated for each neurotransmitter pair to determine the relationship between each neurotransmitter (GraphPad Prism, version 9.1.3).

**Application of machine learning and iterative random forest (iRF) analysis to neurotransmitter data.** Simple correlation-based neurotransmitter networks rely on Pearson's correlation and does not evaluate either nonlinear relationships or causality between neurotransmitter levels for different target classes. We used a state-of-the-art iterative random forest (iRF)[77] regression model called iRF-LOOP (LOOP stands for leave one out prediction)[78,79] that predicts every neurotransmitter level as a function of all other neurotransmitter levels and provides input feature importance in performing the regression. We applied the iRF-LOOP method on every target group (from ['Control','Acute','Chronic']) and concentration (from ['4 mM','30 mM','60 mM','120 mM']) using 80% of data for training and 20% of data for testing the model accuracy. This results in 12 directional PFC neurotransmitter-networks (4 per target class) as depicted in Fig. 8. We also embed the derived feature importance by changing edge thickness and darkness in the networks, i.e., thicker and darker connections are more important, and apply a lower threshold of 0.2 on that to highlight stronger connections (Fig. 8). In addition, using linear regression techniques, we conducted further analysis to determine whether an increase in one neurotransmitter promotes the increase in the levels of another neurotransmitter or suppresses the levels of another neurotransmitter. These are reflected in Fig. 8a as '+' and '−' signs. Finally, we utilize a statistical hypothesis test called Granger Causality[80] to understand the causal impact of GCR classes for all the subjects on K +-evoked time series/and neurotransmitter time series (A) on another neurotransmitter time series (B). The test provides insights into whether knowing the history of A improves the understanding to forecast B beyond that which is possible by just knowing the history of B. The test establishes a causal connection A→ B if a $p$-value of <0.05 is achieved. We set a parameter called 'max-lag' to 3 for the test and used the time series corresponding to all the study subjects to generate the distribution of $p$-values across the GCR classes.

**Reporting summary**
Further information on research design is available in the Nature Portfolio Reporting Summary linked to this article.

## Data availability
All additional data in the manuscript or the supplementary material are available from the corresponding author upon request. Correspondence and requests for materials should be addressed to Rajeev I. Desai. Source data are provided with this paper.

## Code availability
We used open-source Python packages such as Numpy (1.20.3), Matplotlib (3.5.0), pandas (1.4.1), iRF (https://github.com/Yu-Group/iterative-Random-Forest) and Networkx (2.6.3) to analyze and visualize the data. Our codes are available through GitHub: https://github.com/subhamoysgit/neurotransmitter_networks. Correspondence and requests for additional materials should be addressed to Rajeev I. Desai.

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

## Acknowledgements
This study was supported in part by National Aeronautics and Space Administration (NASA), NASA Johnson Space Center grant TXS0147017 (R.I.D.) and NASA NSCOR grant number NNX15AI22G (C.L.L.). M.A.D. Acknowledges support from the NASA SHIELD DRIVE Science Center in BU, grant 80NSSC20K0603.

## Author contributions
R.I.D. conceived and designed the overall study, supervised and analyzed the microdialysis data, and wrote the first draft of the manuscript. B.D.K. conceived and contributed to the design of the neurocognitive study, analyzed the data from the touchscreen experiments, and contributed to the writing of the manuscript. O.T.L. conducted and analyzed the touchscreen experiments. E.S. and E.C.S. ran microdialysis samples in LC-MS/MS and contributed to analyzing the microdialysis data. M.H.D. conducted microdialysis experiments. X.M. and A.M. provided expert LC-MS/MS analytical support for quantification of neurotransmitters. S.C., M.A.D., A.M.-J., and M.I.D. conducted Pearsons correlation analysis, the iterative random forest (iRF) based regression model, and the Granger Causality test for neurotransmitter network analysis. C.L.L. conceived and contributed to the design of the study and the writing of the manuscript. R.I.D. and C.L.L. obtained funding from NASA to conduct the study.

## Competing interests
The authors declare no competing interests.
