## [Peer Review File · Nature Communications]

Complex 33-beam simulated galactic cosmic radiation exposure impacts cognitive function and prefrontal cortex neurotransmitter networks in male miceREVIEWER COMMENTS

Reviewer #1 (Remarks to the Author):

The current manuscript describes the effects of acute and chronic exposure to similar doses of the 33-beam simulated galactic cosmic radiation exposure on various behavioral endpoints, in addition to elicited neurotransmitter responses in the prefrontal cortex. The manuscript details new data regarding neurotransmitter responses following simulated space radiation exposure, which is an addition to field, given that most data focus on terminal measurements of protein expression levels or electrophysiological changes in brain slices. In addition, the authors assessed two different behavioral tests following radiation exposure in the same animals, which provides a method to correlate changes in behavioral performance to changes in neurotransmitter responses. Overall, the manuscript is well-written. However, some conclusions regarding the vigilance test and neurotransmitter studies need to be reevaluated or require additional details.

The main behavioral test that authors are presenting is a psychomotor vigilance test where mice are trained to respond on a lighted square area on a touchscreen. Following a correct response, mice receive a milk reinforcer, in addition to auditory and visual stimuli associated with reinforcer delivery. This test is presented as binary, where mice can either correctly respond or omit a response. There does not seem to be an opportunity for mice to respond prematurely, respond incorrectly, or to exhibit lapses in attention, which are fundamental measures to tests of vigilant attention. Thus, the vigilance aspect of this test is not clear from the description of the test or the data provided in the manuscript. The intermittency of the stimulus varies from 15s – 45s, but this doesn't seem to be a part of the task where the authors measure any responses, such as premature responding. However, the stimulus duration could be as long as (or longer than, apparently) 10 sec, which makes it difficult to understand what aspect of 'vigilance' mice are exhibiting when performing on this test, since the target stimulus is not brief, nor is it delivered in such a way where animals are prepared and waiting for its presentation (e.g., preparatory cues, responding in the magazine to start a trial). From the data that is presented, the mice appear to be demonstrating acquisition of a new touchscreen response; by the end of the training period, all subjects regardless of radiation exposure show discrimination of the touchscreen square stimulus and reliably respond to its occurrence. The parameters described are used commonly to have animals acquire various tests of operant responding, including other published vigilance tests with rodents, but these other tests then go on to further manipulate the response contingencies to assess vigilant attention. Thus, these deficits appear to simply be deficits in acquisition of this test that do not appear to be a function of changes in value of the reinforcer. The 'improvement in vigilance' the authors argue for in the discussion appears to be differences in acquisition, but the authors don't provide any acquisition criteria, so this can't be evaluated. It appears these animals are simply learning this new stimulus-response-outcome association and radiation slows initial responses under these contingencies. The additional data and other clarifications noted below are needed to fully evaluate these behavioral changes.

Titration reaction times – what exactly is this? Are the authors measuring the actual response latency from the time the stimulus duration is illuminated until each mouse makes a correct response (e.g., 10sec stimulus duration, mouse responds at 3.5 sec after stimulus onset, RT recorded as 3.5 sec)? Or is this 'reaction time' simply the stimulus duration at which the mouse responded correctly (e.g., stimulus duration of 10sec, mouse responds correctly, mouse RT is recorded as 10sec)? The actual definition of this measure is not described anywhere in the manuscript. The authors need to present the final average stimulus duration and/or the average stimulus duration for each of the days of testing (only if the titrated reaction time is a true response latency for each animal, and not indicative of the stimulus duration). If the data presented is stimulus duration, why are true response latencies not measured?

Theoretically, a mouse could respond on every trial and by trial ~40 could achieve a stimulus duration of zero – how did the authors handle this? Is there a lower limit to the stimulus duration that once reached, it did not decrease further? Is there an upper limit to the stimulus duration? More details are needed.

The authors need to clearly outline the number of correct trials (they state 48 trials, but do not delineate if this is hits + misses or only hits).

The authors need to present the number of misses (omissions) for each group, since it appears that omissions drive the acquisition deficit or present data regarding correct responses.

The authors need to comment on the lack of a measure of premature responses, or if they have data for premature responding, they need to present it.

The authors need to address how moving the stimulus among the 6 areas from trial to trial includes a measure of divided attention (like the 5-choice RT test); tests can be vigilance tests (sustained attention) or more complex (sustained and divided attention). It seems the authors are presenting a test that could include divided attention. The authors also need to comment on whether or not mice made incorrect responses - i.e., responding on the wrong area of the touchscreen and if these responses were recorded.

The authors should remove the data presented in Figure 3c – it is redundant.

An important aspect to this paper is the lack of behavioral differences between the dose rates, but dose rate effects on the neurotransmitter measurements. The authors need to address this discrepancy. Would they expect similar behavioral differences with different neurotransmitter changes? Are their

behavioral tests just not sensitive enough to pick these up? Would they anticipate dose rate differences for other cognitive domains?

Reviewer #2 (Remarks to the Author):

In this paper Desai and colleagues investigated the effects of acute or chronic exposure to 33-beam GCR on mice behavior and attempted to link behavioral aberrations to changes in neurotransmitter concentration and neurochemical organizational patterns in PFC. I was very excited reading this paper as the topic is generally of importance and quite intriguing.

I will focus my review on my major concerns and leave out any minor comments associated with typos etc.

The idea of the paper is great and the focus would provide new insights. However, I have a number of concerns that I would recommend to be addressed prior to a potential publication:

1) A critical aim of this study is to characterize neurochemical alterations in PFC caused by GCR and link behavioral changes to such alterations. I have a hard time understanding the rationale of the order of experiments. Waiting 8-12 months post-exposure in mice that were already adult, makes me wonder what the neurotransmitter values are actually presenting. Are the indicated differences the result of behavioral training, age or GCR? One may have in fact discovered real effects if the microdialysis was conducted right after the exposure. The neurotransmitter levels in ~18month old mice, which is very advance age are potentially not representative for the effects anymore. The observed effects are in the best case a representation of the convolution of aging, behavioral training and GCR. Analysis of such phenomena is different from the methods presented in this paper.

Moreover, if the study is about figuring out nt changes during a mission, then why were the nt changes not recorded either during the GCR stimulation or right after it. Instead basically when the animals are old and long after the exposure.

2) The authors use 4mM K⁺-stimulation as their chosen method to measure baseline NT concentrations. This is factually not baseline. Even if we agree to consider these values as close enough to a baseline level, the significant differences in these "baseline" concentrations between control, acute and chronic

group, make the interpretation of the analysis of higher level stimulations questionable. Dynamical systems are sensitive to initial values. Differences in initial values (even small ones) can lead to non-linear differences in response to stimulations and this is very likely in case of neurotransmitters. Consequently, once baselines are significantly different, the stimulated levels are not necessarily comparable anymore.

3) Page 9 line 180: The authors state that glu and gaba were not significantly different from baseline. This contradicts Table 1.

4) Page 9 lines 181-182: This sentence is not clear.

5) reorganization of neurochemical networks:

5a) what is the meaning of correlation between neurotransmitter concentrations following a challenge? Why not looking into causality and coherence analysis?

5b) This figure and the numbers are quite worrisome: Any value between -0.5 and 0.5 is not very meaningful. The EIN and glu-GABA interactions in PFC are well-known. What is worrisome though, is that this major network is in this analysis negatively (and majorly) affected by the K⁺ stimulation. This observation is not explained, namely why higher k⁺ stimulation, has led to vanishing correlation within EIN. Moreover, the time-scales of microdialysis experiments, makes such correlations for interpretation of local changes not very appropriate. Sentence 233 is furthermore not accurate.

5c) applications of high k⁺ concentrations (100 mM) have been shown to induce distinct oscillations in DA levels. I am wondering why the authors increased the K⁺ stimulation to 120mM but didnt discuss the consequences of such stimulation and that this has not much to do with behavioral patterns under normal conditions. Simply arguing that "the temporal patterns of other neurotransmitters were less affected, requiring greater local K⁺-

362 stimulation to observe any differences", is not sufficient. Very high stimulations lead to very different dynamics, it is not necessarily linear.

6) Page 15 line 316: The authors state "Likewise, we found that acute or chronic delivery of the 33-beam GCR generally produced similar and less severe neurocognitive and electrophysiological impairments in rodents". In this paper, there is not a single electrophysiological experiment or even statement. Even if the authors were partly involved in the other study [34], this sentence needs to be rephrased. Otherwise, it is very misleading.

7) Page 17 line 354: The authors mention that they results with respect to glu-gaba system, which is the most critical system within cortical structures, is not in agreement with previous studies. How do the authors justify this?

8) Page 18 line 384: This interpretation is far over-fetched and not really clear. The corr coeffs are so low that there is not real meaning in it. Moreover, it is not clear why EIN is lost in baseline case.

9) Page 36 online methods: There are certain standards of reporting methods that are not followed. number of animals is not provided. Origin of the animals is not clear. Currently, it is a copy-paste ("Jackson Laboratories, City, State"). The statement about "~6-9 months old (~20-25 years in human age)" is not meaningful. It is not discussed why no female animals were used although many studies indicate sex-specific effects.

In summary, the topic of this study is interesting but in my humble opinion, the choice of methods and the findings do not fully justify the main conclusion of this study.

Reviewer #3 (Remarks to the Author):

Desai and collaborators (ms#NCOMMS-23-06284-T) study a fascinating topic regarding the potential impact of galactic cosmic rays (GCR) exposure during space exploration in cognitive behaviors and neurochemistry. For this purpose, authors simulated GCR in the lab by acutely or chronically exposing adult C57BL/6 mice to a complex 33-beam composition of GCR. Observations were compared to control mice not exposed to GCR but otherwise underwent similar handling and experimental procedures. GCR exposed mice were then behaviorally tested using automated touchscreens. Neither acute nor chronic GRC exposure affected reward sensitivity during an Economic Demand Task but increased reaction times to respond to stimuli in a Psychomotor Vigilance Task. Finally, in vivo microdialysis experiments recorded extracellular baseline and KCl-evoked levels of dopamine, GABA, glutamate, serotonin (5-HT), and norepinephrine in the prefrontal cortex (PFC). Authors report that acute and/or chronic exposure to GCR increased basal and KCl-evoked levels of all the neurotransmitters tested when compared to levels in control mice. Finally, authors used a correlational and machine learning approach to study whether changes in the GCR-induced disbalance in PFC impacted neurotransmitter networks.

Major comments:

Overall, experiments and data presented here seem well performed. The major criticism is however that this manuscript lacks a rationale/story line that helps the reader to understand the selection of the experiments performed, and why they are relevant in the conclusions here depicted. For instance, it is

not fully clear why a behavioral task to assess reward sensitivity was used along an attentional task if the main scope was to test for potential cognitive deficits related to exposure of GCR. Was this the only goal in this study? Furthermore, why authors decided to test for neurotransmitters in the PFC alone, but not from other brain regions also directly or indirectly associated to cognitive function? What was the rationale to study only the above-mentioned neurotransmitters and not others? Why is relevant a correlation analysis between pair of neurotransmitters? Did the authors consider any functional criteria (e.g., biosynthetic pathways, excitability, neuromodulatory, etc.) to match and assign different weights of functional relevance? How this analysis helps to understand neurochemical networks in the PFC?

Additionally, it is hard to connect the significance of the different datasets presented in this manuscript into a single major conclusion. For example, why GCR affected behavior and extracellular neurotransmitters in the PFC? what is the relationship between an increased reaction time to touch stimuli in GCR-exposed mice and the abnormal neurochemical fingerprint in PFC? What do we learn by using a KCl-evoked response that cannot be concluded from the baseline levels alone? How altered neurochemical networks in the PFC may explain an abnormal behavioral phenotype?

Economic Demand task: Unclear why authors are comparing performance following different reward concentrations (50%, 20%, 5% or 0% milk concentration) and different responding, all throughout the same task. Is there evidence that this is a measure of motivation? More discussion of the rationale for using this task and the dependency of it on the PFC is needed.

Psychomotor Vigilance Task: Why choose 20% concentration for this task following the variable reward concentrations of previous task? Explain. Acute and chronic exposure increased response times. Is this a real attentional impairment? Is there evidence that this not a learning impairment (as the reaction times are most effected early in training) or a motor impairment?

Minor comments:

- Abstract: why authors mention only dopamine but not the other neurotransmitters tested?
- Introduction: The study of PFC circuits – it is mentioned in the introduction that neurocognitive performance should be studied, but the rationale for studying PFC is lacking.
- Must be more clear what sexes you are studying (only males) and why. In the discussions, authors specify the role of sex in similar work by Dr. Amelia Eisch, but do not discuss this further in relations to their own findings.
- Age of mice: on line 726: specifies that they used 6-9 months of age mice which is equivalent to 20-25 years of human age. According to Jackson laboratory, 6-9 months of age would be equivalent to 30-40 years of human age
- Graphs should have SEM or individual variability presented for readers interpretation of results (Fig 2c).
- Line 43: Typo 'prefrontal'
- Line 170: Typo 'Neurotransmitters'
- Lines 195, 197, 204, 211, 214, 217, 219: please provide units to data presented

- Lines 734-745 (irradiation protocol): Could you please comment why acute vs. chronic exposure radiation used different times (2 hours vs. 80 minutes)? Also, why animals were restrained \approx 20 mins (line 745) if radiation times were longer?
- Line 850: What was the molecular weight cut-off of your dialysis probe?
- Lines 867-869: why did you use 3 different concentrations of KCl?
- Figure 4c: please provide specific probe location instead of overall region.
- Figure 7b and 8b: figures seem stretched in the vertical plane. Please correct.

Response To Reviewers:

Reviewer #1:

1) The current manuscript describes the effects of acute and chronic exposure to similar doses of the 33-beam simulated galactic cosmic radiation exposure on various behavioral endpoints, in addition to elicited neurotransmitter responses in the prefrontal cortex. The manuscript details new data regarding neurotransmitter responses following simulated space radiation exposure, which is an addition to field, given that most data focus on terminal measurements of protein expression levels or electrophysiological changes in brain slices. In addition, the authors assessed two different behavioral tests following radiation exposure in the same animals, which provides a method to correlate changes in behavioral performance to changes in neurotransmitter responses. Overall, the manuscript is well-written. However, some conclusions regarding the vigilance test and neurotransmitter studies need to be reevaluated or require additional details.

The main behavioral test that authors are presenting is a psychomotor vigilance test where mice are trained to respond on a lighted square area on a touchscreen. Following a correct response, mice receive a milk reinforcer, in addition to auditory and visual stimuli associated with reinforcer delivery. This test is presented as binary, where mice can either correctly respond or omit a response. There does not seem to be an opportunity for mice to respond prematurely, respond incorrectly, or to exhibit lapses in attention, which are fundamental measures to tests of vigilant attention. Thus, the vigilance aspect of this test is not clear from the description of the test or the data provided in the manuscript. The intermittency of the stimulus varies from 15s – 45s, but this doesn't seem to be a part of the task where the authors measure any responses, such as premature responding. However, the stimulus duration could be as long as (or longer than, apparently) 10 sec, which makes it difficult to understand what aspect of 'vigilance' mice are exhibiting when performing on this test, since the target stimulus is not brief, nor is it delivered in such a way where animals are prepared and waiting for its presentation (e.g., preparatory cues, responding in the magazine to start a trial). From the data that is presented, the mice appear to be demonstrating acquisition of a new touchscreen response; by the end of the training period, all subjects regardless of radiation exposure show discrimination of the touchscreen square stimulus and reliably respond to its occurrence. The parameters described are used commonly to have animals acquire various tests of operant responding, including other published vigilance tests with rodents, but these other tests then go on to further manipulate the response contingencies to assess vigilant attention. Thus, these deficits appear to simply be deficits in acquisition of this test that do not appear to be a function of changes in value of the reinforcer. The 'improvement in vigilance' the authors argue for in the discussion appears to be differences in acquisition, but the authors don't provide any acquisition criteria, so this can't be evaluated. It appears these animals are simply learning this new stimulus-response-outcome association and radiation slows initial responses under these contingencies. The additional data and other clarifications noted below are needed to fully evaluate these behavioral changes.

We appreciate the critical review of our procedural details regarding this variant of the psychomotor vigilance task. As the reviewer correctly notes, our touchscreen-based variant of the psychomotor vigilance task departs in some ways from traditional rodent versions. These departures, we believe, have both pros and cons. Our task variant is largely based on "radar" task procedures from human studies where the subject attends to a

screen for blips on the radar and must respond to them following various intermittencies in stimulus presentation. We have used this variant previously in rodents and nonhuman primates to successfully examine various drug effects on attentional processes (e.g., Kangas et al., 2016, PMID 26826191; Johnson et al., 2021, PMID 33712507). As the reviewer correctly notes, there is indeed a binary choice where the mouse must closely attend to the screen and respond quickly to obtain reward. Because the stimulus appears on the screen in different areas and then disappears if no response occurs during the titrated stimulus duration interval, subjects can either respond to the stimulus accurately or omit/miss a response. Premature responses are not possible because the subject must attend to the entire screen and is not sure where the stimulus will be presented during the next trial. As the reviewer notes, premature responses are a very common and important datum of the PVT task when using fixed operanda (e.g., nose poke holes or levers) but that is not a feature of this touchscreen variant. Regarding the reviewer's other point, subjects can indeed exhibit lapse in attention, which would be evident in missed responses across trials and would in turn increase the duration of stimulus presentation on the subsequent trial. We have now added additional detail in the text regarding these procedural differences (see Online Methods section).

Apologies for the confusion regarding the intermittency of the stimulus. Following sufficient exposure to the task, the target stimulus is indeed brief and made more and more difficult as the subject continues to master the task. The stimulus appears on the screen for up to 10 sec on the first trial of the first session, but that duration is made 0.25 sec briefer after every correct response (and made 0.25 sec longer after each miss). These titration contingencies serve to "dial in" the attentional abilities of the subject to determine, in this case, group differences in GCR groups. The subjects are, as the reviewer says, prepared and waiting for stimulus presentation, and vigilance in this case is exhibited by the necessary attention required to scan the area of the touchscreen, while not knowing where (on the screen) or when after the last trial (either 15, 30, or 45 s) the stimulus will appear. As referenced above, although a departure from some other rodent PVT variants, this is designed after the radar studies in humans. Again, we regret the confusion and hope our manuscript revisions clear things up.

Regarding deficits in acquisition of this test. It is true that in addition to examining potential differences among each treatment group in their ability to maintain brief titrated reaction time intervals, we were also interested in potential group differences in the *development* of this task performance. As shown in Figure 3, both the development of task performance (i.e., how many sessions it took to display asymptotic performance) as well as what level that asymptotic performance plateaued, were affected by GCR treatment. Since these studies were designed to obtain as much information as possible from these valuable subjects, we did not run either the PVT or demand task to a predefined stability criteria, but rather for a fixed number of sessions (15 sessions for the PVT) to allow them to continue to *in vivo* microdialysis studies in a within-subject manner while still in middle adulthood (i.e., astronaut-aged).

2) Titrated reaction times – what exactly is this? Are the authors measuring the actual response latency from the time the stimulus duration is illuminated until each mouse makes a correct response (e.g., 10sec stimulus duration, mouse responds at 3.5 sec after stimulus onset, RT recorded as 3.5 sec)? Or is this 'reaction time' simply the stimulus duration at which the mouse responded correctly (e.g., stimulus duration of 10sec, mouse responds correctly, mouse RT is recorded as 10sec)? The actual definition of this measure is not described anywhere in the manuscript. The authors need to present the final average stimulus duration and/or the average stimulus duration for each of the days of testing (only if the titrated reaction time is a true response latency for each animal, and not indicative of the stimulus duration). If the data presented is stimulus duration, why are true response latencies not measured?

The reviewer makes an important point. In response, we have reanalyzed *all PVT data* in accord with it. In the previous version, our "Titrated Reaction Time" metric was the average *programmed* stimulus duration. For example, in the first trial of the first session, the stimulus duration value was set to 10 sec. So, the stimulus appeared on the screen until it was touched or until it disappeared 10 sec later. If touched, the duration of presentation on the next trial was reduced by 0.25 sec; if it was missed, it was increased by 0.25 sec. And, as stated in the Method section, the stimulus duration value from the last trial of one session would be the value of the first trial of the next session. Thus, what was plotted was the average titrated duration of stimulus presentation.

That said, after considering the reviewer's comments, we agree that a better dependent measure should be the actual reaction time measure of the subject under these conditions. Therefore, we have reanalyzed all of the PVT data and now plot the session means of their reaction times under these titrating contingencies of stimulus presentation. As shown in the revised Figure 3 (and associated revised manuscript text), this reanalysis does not change the overall findings or interpretations. In fact, the shape of the functions are highly similar and, as expected, shifted downward now that true reaction times (response latencies) are plotted. Nevertheless, it is a more descriptive measure of performance. We are grateful to the reviewer for suggesting this.

3)Theoretically, a mouse could respond on every trial and by trial ~40 could achieve a stimulus duration of zero – how did the authors handle this? Is there a lower limit to the stimulus duration that once reached, it did not decrease further? Is there an upper limit to the stimulus duration? More details are needed.

There was no upper limit, but the lower limit was set to 0.25 sec to avoid a stimulus duration of zero on a given trial. This detail is now included in the Method (see Online Methods section).

4)The authors need to clearly outline the number of correct trials (they state 48 trials, but do not delineate if this is hits + misses or only hits). The authors need to present the number of misses (omissions) for each group, since it appears that omissions drive the acquisition deficit or present data regarding correct responses.

For both reviewer points, we believe this is now rectified following our reanalysis of data. That is, per the reviewer recommendation above, we now present the actual mean reaction time rather than mean programmed stimulus duration.

5)The authors need to comment on the lack of a measure of premature responses, or if they have data for premature responding, they need to present it.

As we describe above, this touchscreen-based variant of the PVT does not allow for premature responses (in the traditional sense) because there are no fixed operanda to respond prematurely on. We did not record responses to a blank screen when stimuli were not presented because we believe making conclusions based on blank screen responses would be difficult. In the course of the manuscript revision, we realized that the previous version of the task schematic on Figure 3 had white outlines to indicate the potential stimulus positions. However, this was for illustrative effect, as these white lines were not actually presented on the touchscreen. Therefore, we have removed them to avoid confusion. See the Online Methods section for related manuscript revisions regarding the missing premature response metric from this task variant.

6)The authors need to address how moving the stimulus among the 6 areas from trial to trial includes a measure of divided attention (like the 5-choice RT test); tests can be vigilance tests (sustained attention) or more complex (sustained and divided attention). It seems the authors are presenting a test that could include divided attention. The authors also need to comment on whether or not mice made incorrect responses - i.e., responding on the wrong area of the touchscreen and if these responses were recorded.

The reviewer makes a highly interesting comment. We agree that because this PVT task variant has the stimuli appearing randomly among 6 different positions it is likely capturing some features of the 5CSRT task, perhaps divided attention. However, as we detail above and in the revised manuscript, unlike the 5CSRT task and traditional rodent PVT, there are no fixed operanda. Rather the stimuli appear and disappear from the screen. And because we didn't view touches to various parts of a blank screen as incorrect (or premature) responses, we did not record such activity.

7)The authors should remove the data presented in Figure 3c – it is redundant. As the reviewer recommends, we have now removed Figure 3c.

8) An important aspect to this paper is the lack of behavioral differences between the dose rates, but dose rate effects on the neurotransmitter measurements. The authors need to address this discrepancy. Would they expect similar behavioral differences with different neurotransmitter changes? Are their behavioral tests just not sensitive enough to pick these up? Would they anticipate dose rate differences for other cognitive domains?

As the reviewer may be aware, in the field of radiobiology dose rate has a long history, and a wealth of literature has demonstrated that lowering the dose rate from Gy/min to cGy/hr can ameliorate many adverse effects due to the temporal superposition of DNA repair during dose delivery. That data has traditionally relied on cell kill data derived from cancer cells and does not remotely reflect how functional endpoints in the brain respond to different dose rates. Importantly, this study was not designed to discriminate differences in dose rate – but rather to elucidate how a complex 33-beam GCR simulation might impact CNS functionality at the behavioral and neurochemical level. For the actual chronic irradiations, dose rates for individual energetic particles varied between 0.04 and 0.9 cGy/min – with a collective mean dose rate of $\sim 2.08 \text{ cGy/1h} = 0.035 \text{ cGy/min}$ or 0.35 mGy/min . For the actual acute irradiations, dose rates for individual energetic particles varied between 0.16 and 4 cGy/min – with a collective mean dose rate of $50 \text{ cGy/2h} = 0.42 \text{ cGy/min}$ or 4.2 mGy/min . As such, unlike traditional assessments of dose-response function in pharmacological studies, our behavioral tasks are simply not sensitive enough or designed to pick up this small difference in radiation dose rates. We reiterate that the focus of this study was to evaluate beam complexity (mixed fields of energetic particles) not dose rate. We also point out that very few studies have systematically and rigorously evaluated the impact of space relevant dose rates on CNS functionality. The one study that did (Acharya et al., 2019, PMID 31383727) implemented a dose rate of 1 mGy/day over the course of six months, and reported an array of adverse behavioral, electrophysiological, and molecular outcomes compared to unirradiated controls. Portions of the foregoing have been added to the discussion at your request (see pg. 15).

Reviewer #2:

1) In this paper Desai and colleagues investigated the effects of acute or chronic exposure to 33-beam GCR on mice behavior and attempted to link behavioral aberrations to changes in neurotransmitter concentration and neurochemical organizational patterns in PFC. I was very excited reading this paper as the topic is generally of importance and quite intriguing.

I will focus my review on my major concerns and leave out any minor comments associated with typos etc. The idea of the paper is great and the focus would provide new insights. However, I have a number of concerns that I would recommend to be addressed prior to a potential publication:

We appreciate the kind words and are likewise excited about the findings reported in the manuscript.

2) A critical aim of this study is to characterize neurochemical alterations in PFC caused by GCR and link behavioral changes to such alterations. I have a hard time understanding the rationale of the order of experiments. Waiting 8-12 months post-exposure in mice that were already adult, makes me wonder what the neurotransmitter values are actually presenting. Are the indicated differences the result of behavioral training, age or GCR? One may have in fact discovered real effects if the microdialysis was conducted right after the exposure. The neurotransmitter levels in ~ 18 month old mice, which is very advance age are potentially not representative for the effects anymore. The observed effects are in the best case a representation of the convolution of aging, behavioral training and GCR. Analysis of such phenomena is different from the methods presented in this paper.

Moreover, if the study is about figuring out nt changes during a mission, then why were the nt changes not recorded either during the GCR stimulation or right after it. Instead basically when the animals are old and long after the exposure.

We appreciate the critical appraisal of the timeline/order of our performed studies as well as the opportunity to respond/clarify. The overall design/order of our experiments was based on both the scientific and practical considerations highlighted below.

As the reviewer may be aware, there is limited information on how exposure to a complex 33-beam GCR simulation impacts translational touchscreen behavior and brain neurochemistry in freely moving subjects. As such, our overarching goal was to obtain as much information as possible on these endpoints from the limited number of valuable subjects that were available. We strongly believe that generating within-subject data on neurotransmitter changes in the same group of animals that performed touchscreen behavioral task represents a powerful approach that allows us to meaningfully relate changes in PFC neurotransmitter networks (albeit at specific time point) to behavioral/neurocognitive consequence compared to other between-subjects study designs (also see below response as well as response to reviewer 2, point 11 and reviewer 3 point 5).

The reviewer correctly makes an important and thought-provoking point regarding the other factors (e.g., behavioral training and age) in addition to GCR exposure that could also contribute to the overall observed PFC neurotransmitter network changes reported in this manuscript. We carefully considered these issues prior to designing/conducting these studies. We would like to emphasize that while assessment of age and behavioral training history were not part of the overall study goals, it is noteworthy that the animals in the control group were of the same age with the comparable behavioral history as the GCR exposed animals, meaning that while age and behavioral training may be important variables, it is one that we considered in the experimental design. Nevertheless, we appreciate the reviewer's point that the results presented may be a consequence of how these controlled variables may interact with the neurotransmitter changes elicited by GCR simulation (see also point 5 response). In the revised manuscript, we have added text to the discussion that acknowledges these considerations (see page 19-20).

We agree with the reviewer's other point that probing neurotransmitter network changes in the PFC at multiple time points (e.g., early, middle, and late) after GCR exposure may reveal further insights into how *in vivo* neurotransmitter network dynamics are altered over time to impact behavior. Unfortunately, such an endeavor would require a substantially larger cohort of subjects that were simply not available and therefore was beyond the scope of this work. Nevertheless, it is noteworthy, that our data showing long-lasting impact of acute and chronic GCR exposure on behavior and neurotransmitter consequences is highly consistent with the long-term adverse effects on other behavioral and molecular endpoints that have been documented following exposure to space radiation (References 7-12, 21-24). We strongly believe that the data presented here sets a solid benchmark for an expanded evaluation of *in vivo* neurotransmitter network dynamics over time following GCR exposure. From a broader perspective, the reviewer correctly points out that the neurotransmitter network reorganization may be different immediately after GCR exposure vs at later time points. Thus, it is possible that the detrimental effects of both acute or chronic GCR on both touchscreen task and neurotransmitter function may have dissipated and might be even more pronounced early after GCR exposure. However, in the context of a long duration spaceflight, the changes we measure are likely more reflective of the accumulated GCR dose and acute changes (if they do occur) have not been documented and would not translate to late onset radiation-induced cognitive decrements known to take time (months) to manifest after such low dose complex radiation exposures. We therefore contend that our protracted measurements are much more relevant to NASA risk assessments where potential decrements are more likely to manifest on the return from Mars rather than en route to Mars. Thus, while late changes in neurotransmitter levels may reflect the combined impact of all experimental stressors, under our experimental conditions, we believe these are most likely the result of permanent radiation-induced changes to the basal neurochemical state of the irradiated brain. In the revised manuscript, we have added text to the discussion that acknowledges these considerations (see page 21-22).

From a practical perspective, as the reviewer maybe aware, collection of microdialysis samples over time (every 20 min for up to 320 min) in freely moving animals during ongoing GCR simulation is technically not feasible due to safety concerns and limitations at NASA Space Radiation Laboratory (BNL, NY) facility.

Finally, the reviewer's point regarding conducting microdialysis studies immediately after exposure is well taken. However, given our overall goals, touchscreen behavioral assays needed to precede microdialysis studies as all subjects were sacrificed immediately following neurotransmitter sample collection to verify probe placements in the PFC. Please note that we and others have used this type of within-subject approach in prior work to demonstrate meaningful relationships between pharmacologically induced neurotransmitter changes and complex behavior (e.g., Desai et al., 2010; Puhl et al., 2019).

3) The authors use 4mM K⁺-stimulation as their chosen method to measure baseline NT concentrations. This is factually not baseline. Even if we agree to consider these values as close enough to a baseline level, the significant differences in these "baseline" concentrations between control, acute and chronic group, make the interpretation of the analysis of higher level stimulations questionable. Dynamical systems are sensitive to initial values. Differences in initial values (even small ones) can lead to non-linear differences in response to stimulations and this is very likely in case of neurotransmitters. Consequently, once baselines are significantly different, the stimulated levels are not necessarily comparable anymore.

We appreciate the reviewer's point. In the revised manuscript, we now further clarify (page 47-48, Online Methods section) that the ringer solution consisting of 4 mM K⁺ as well as NaCl and CaCl₂, is typically considered to be comparable to artificial cerebrospinal fluid in microdialysis studies and has been used, with some minor variations (2.7–4.8 mM K⁺), by us (e.g., Desai et al., 2010; Fisher et al., 2018; Puhl et al., 2019) and others (see below) for decades as the standard perfusate for establishing *in vivo* baseline levels of neurotransmitters in studies assessing effects of various stressors (environmental, age, pharmacological). Below, we provide a selection of publications to support this:

1. Jackson, M. E., Frost, A. S., & Moghaddam, B. (2001). Stimulation of prefrontal cortex at physiologically relevant frequencies inhibits dopamine release in the nucleus accumbens. *Journal of Neurochemistry*, 78(4), 920–923. <https://doi.org/10.1046/j.1471-4159.2001.00499.x>
2. Käenmäki, M., Tammimäki, A., Myöhänen, T., Pakarinen, K., Amberg, C., Karayiorgou, M., Gogos, J. A., & Männistö, P. T. (2010). Quantitative role of COMT in dopamine clearance in the prefrontal cortex of freely moving mice. *Journal of Neurochemistry*, 114(6), 1745–1755. <https://doi.org/10.1111/j.1471-4159.2010.06889.x>
3. Kihara, T., Ikeda, M., Miyazaki, H., & Matsushita, A. (1993). Influence of Potassium Concentration in Microdialysis Perfusate on Basal and Stimulated Striatal Dopamine Release: Effect of Ceruletide, a Cholecystokinin-Related Peptide. *Journal of Neurochemistry*, 61(5), 1859–1864. <https://doi.org/10.1111/j.1471-4159.1993.tb09827.x>
4. Perry, K. W., Falcone, J. F., Fell, M. J., Ryder, J. W., Yu, H., Love, P. L., Katner, J., Gordon, K. D., Wade, M. R., Man, T., Nomikos, G. G., Phebus, L. A., Cauvin, A. J., Johnson, K. W., Jones, C. K., Hoffmann, B. J., Sandusky, G. E., Walter, M. W., Porter, W. J., ... Svensson, K. A. (2008). Neurochemical and behavioral profiling of the selective GlyT1 inhibitors ALX5407 and LY2365109 indicate a preferential action in caudal vs. Cortical brain areas. *Neuropharmacology*, 55(5), 743–754. <https://doi.org/10.1016/j.neuropharm.2008.06.016>
5. Ripley, T. L., Jaworski, J., Randall, P. K., & Gonzales, R. A. (1997). Repeated perfusion with elevated potassium in *in vivo* microdialysis—A method for detecting small changes in extracellular dopamine. *Journal of Neuroscience Methods*, 78(1), 7–14. [https://doi.org/10.1016/S0165-0270\(97\)00129-](https://doi.org/10.1016/S0165-0270(97)00129-)
6. Sanna, F., Bratzu, J., Serra, M. P., Leo, D., Quartu, M., Boi, M., Espinoza, S., Gainetdinov, R. R., Melis, M. R., & Argiolas, A. (2020). Altered Sexual Behavior in Dopamine Transporter (DAT) Knockout Male Rats: A Behavioral, Neurochemical and Intracerebral Microdialysis Study. *Frontiers in Behavioral Neuroscience*, 14. <https://www.frontiersin.org/articles/10.3389/fnbeh.2020.00058>
7. Mereu M, Tronci V, Chun LE, Thomas AM, Green JL, Katz JL, Tanda G (2015) Cocaine-induced endocannabinoid release modulates behavioral and neurochemical sensitization in mice. *Addict Biol* 20:91–103.

We would also like to emphasize that the baseline values observed at 4 mM K⁺ include the effect of GCR stimulation and do not attempt to establish a generalizable baseline for all animals, rather they establish what GCR vs control groups neurotransmitter levels are as baseline due to the differing conditions. While we agree that high level stimulation values are not directly comparable in isolation, relative to each group's baseline values they express the relative dynamic K⁺-evoked excitability of the system over time, which we believe is comparable between groups at each K⁺ concentration.

4) Page 9 line 180: The authors state that glu and gaba were not significantly different from baseline. This contradicts Table 1.

Apologize for any confusion. The * in Table 1 indicates significant difference between acute or chronic GCR groups vs controls and the # indicates significant difference between acute vs chronic GCR animals. The statement on page 9, line 180 has been modified to more accurately reflect this.

5) Page 9 lines 181-182: This sentence is not clear.

This sentence has been clarified in the revised manuscript (see page 9).

6) Reorganization of neurochemical networks:

6a) what is the meaning of correlation between neurotransmitter concentrations following a challenge? Why not look into causality and coherence analysis?

We thank the reviewer for pointing this out. We wanted to start with a simple analysis to understand the impact of one neurotransmitter state on another and how those change when the acute and chronic GCR exposed subjects are stimulated with different concentrations of K+. As shown in the revised Figure 7a-c, we observed significant reorganizations in the neurotransmitter networks constructed through Pearson correlation between each neurotransmitter-pair. However, Pearson correlation does not tell the entire story as it cannot reflect the combined impact of multiple neurotransmitters on one neurotransmitter. To better appreciate this connection, we constructed the neurotransmitter networks (or graphs) using a *state-of-the-art* method called iRF-LOOP that makes use of iterative Random Forest regression and generates adjacency matrices reflecting instantaneous impacts on one neurotransmitter by the rest (see revised Figure 8a-b). Using this approach, we further confirmed the significant reorganizations in PFC neurotransmitter networks depending on K+ concentration and target GCR class. This approach not only provided the depiction of connection of multiple neurotransmitters but also represented the directions of the impacts. As per reviewer suggestion, in Figure 8 a and b of the revised manuscript, we have now conducted further analysis (linear regression) and added a '+' sign of impact to reflect that an increase in one neurotransmitter promotes the increase in the levels of another neurotransmitter, whereas a '-' sign reflects suppression. It is noteworthy that the above approaches do not treat the input data as time series. Following the reviewer's suggestion, we have now performed a causality analysis to better appreciate the causal impact of the neurotransmitter changes induced by different GCR exposure levels. We performed the Granger Causality test to understand the significance of those causal impacts and generated boxplots depicting the distribution of p-values over all the subjects for DA's impact on each neurotransmitter and vice versa (see Figure 9 in the revised manuscript). We primarily focused on DA, as our data indicate this neurotransmitter to be a key player in the reorganized neurotransmitter networks. We found that for control group, the median p-values are greater than 0.05 for all the neurotransmitters impacting DA and vice versa (Figure 9a and b), implying that the other neurotransmitters do not impact DA and vice versa in the control subjects. However, based on median p-values, in subjects exposed to acute GCR only the neurotransmitter GABA impacts DA (Figure 9a) and in acute GCR group the causal impact of other neurotransmitters (except GABA) on DA cannot be meaningfully established. Moreover, Figure 9 shows that DA in turn impacts GABA, GLU, and 5-HT but not NE. Likewise, in subjects exposed to chronic GCR, the median p-values show that GABA and NE impact DA and vice versa. This Granger Causality test provides a deeper insight into the PFC neurotransmitter network that was clearly altered over protracted acute and chronic post-exposure times. Evident from this analysis is that DA has a significant impact on irradiated neurotransmitter levels (most notably GABA, Fig. 9A), while the other neurotransmitters also impact DA levels (most notably after chronic exposures, Fig. 9B) when compared to controls. The reciprocity of the neurotransmitter network strengthens our arguments for the functional reorganization of these networks following space radiation exposure. While it remains difficult to pinpoint the incipient event that triggered such widespread change, it may also prove relatively unimportant in the context of chronic exposures encountered during extended deep space travel. To the extent that our machine learning and causality analyses are predictive of network level reorganization, then it seems plausible that such long-term alterations will be an inevitable consequence of long duration space travel. What is less certain, is whether such reorganization of neurotransmitter interactions will continue to evolve and if/how they will functionally impact critical decision making and adaptive

reasoning under more stressful situations in space. Future work that includes a larger data set will further strengthen the confidence in the results of the presented analyses. We have now added/modified text throughout the manuscript to include the findings of this analysis. We thank the reviewer for this suggestion.

6b) This figure and the numbers are quite worrisome: Any value between -0.5 and 0.5 is not very meaningful. The EIN and glu-GABA interactions in PFC are well-known. What is worrisome though, is that this major network is in this analysis negatively (and majorly) affected by the K⁺ stimulation. This observation is not explained, namely why higher k⁺ stimulation, has led to vanishing correlation within EIN. Moreover, the time-scales of microdialysis experiments, makes such correlations for interpretation of local changes not very appropriate. Sentence 233 is furthermore not accurate.

The reviewer makes a good point. We used statistical significance (p-value < 0.05) as the primary measure for plotting pairwise interactions between neurotransmitters in the PFC in Figure 7c. In the revised manuscript, we have now modified Figure 7c to show pairwise interactions between neurotransmitters based on values that only include Pearson's correlation ≥ 0.5 with a significance p-value of <0.05 (see Page 11). We have modified the text accordingly in the results and discussion sections (see page 11-12) as well as Supplementary Table 2 to reflect this change. Note that such an approach is in line with previously published work (Zhang et al., 2020). Importantly, while this has eliminated some of the weaker interactions that were previously included in the pairwise network plot, the overarching interpretation that GCR exposure reorganizes PFC neurotransmitter networks, especially the relationship among monoamines, does not fundamentally change. We thank the reviewer for this guidance.

We appreciate the concern posed by the reviewer regarding network correlations and K⁺ stimulation. As illustrated in the response of 6a above, we have now performed a causality analysis to understand how GCR exposure alters the impact of other neurotransmitters on DA and vice versa (see Figure 9 in revised manuscript). As noted in other responses, the K⁺ stimulation is simply a means to study how a particular stressor (i.e., acute vs chronic GCR exposure compared to controls) impacts PFC neurotransmitter dynamics. Our intention is not to study how K⁺ stimulation alters neurotransmitter responses, but how a stressor like GCR alters the K⁺-evoked neurotransmitter responsiveness. Nevertheless, following the reviewer's suggestion, we have now performed a causality analysis to better appreciate the impact of the temporal changes induced by different concentration of K⁺ on each neurotransmitter state and investigate how that causal impact changes with different GCR exposure levels. We performed the Granger Causality test to understand the significance of those causal impacts and generated boxplots depicting the distribution of p-values over all the subjects for each neurotransmitter. As shown below, we find that for control group, the median p-values are less than 0.05 for all the neurotransmitters implying significant causal impact of stimulation with different K⁺ concentrations. However, based on median p-values, the causal impact of K⁺ variation cannot be meaningfully established on NE and GLU levels in 'Acute' group, and on NE levels in 'Chronic' group. This analysis thus reflects significant changes in the response of the PFC neurotransmitter network to external stimulations (K⁺ level) that are dependent on the type of GCR exposure.

We appreciate the reviewer's point that the temporal resolution (i.e., >5 min) for microdialysis procedures may make correlating and interpreting momentary local changes in the PFC challenging. Although, we agree that the correlations presented here cannot be extrapolated to shorter time intervals (e.g., milliseconds or seconds) that can be observed using other neurochemical methods with faster time resolutions (e.g., fast scan cyclic voltammetry; FSCV), we believe that we have appropriately presented/discussed these data with the understanding that they represent larger and more persistent neurotransmitter network level changes than simply brief interactions between neurotransmitter systems. In this regard, we emphasize that microdialysis techniques have been the gold standard for real-time monitoring of steady and persistent neurochemical changes in the field for decades and has two notable advantages compared to other methods: a) the ease with which a variety of small molecule analytes, including the neurotransmitters reported in this manuscript, that can be measured compared to FSCV which requires the target neurotransmitter to be electroactive and distinct from interferents; and b) the ability to deliver solutions to targeted brain regions through the microdialysis probe to induce a physiological change in that region leading to alterations in neurotransmitter dynamics. Here, we leveraged both advantages by analyzing how ongoing probe-delivery of different concentrations of K⁺ solutions over time caused perturbations of the PFC neurotransmitter networks associated with monoamines, Glu, and GABA. Much like generating dose-response functions in pharmacological studies, our approach here permitted the determination of K⁺-evoked concentration-dependent changes in neurotransmitter dynamics over time, which we believe to be a superior method of determining neurotransmitter dynamics compared to assessing changes induced by a single K⁺ concentration. The idea being that increasing K⁺ levels will result in alterations in physiological conditions (e.g., increasing physiological stress) on the PFC neurotransmitter system such that both subtle and obvious dynamic neurotransmitter responses can be easily detected (see also response to reviewer 3, point 5). The importance of this tactic is further highlighted by the observed concentration- and time-dependent increases and decreases in DA levels in the control mice vs GCR exposed animals, i.e., DA levels increased within 20 min of each K⁺ change and declined thereafter and this effect was slightly dampened following each subsequent increase in K⁺-concentration (Figure 6a); whereas the GCR exposed mice could not produce this normal response. These are remarkable data and likely are related to the underlying processes of DA (and other neurotransmitters) synthesis, release, action, and uptake taking minutes rather than milliseconds/seconds. As noted above (see response to reviewer 2, point 3), this type of time resolution and approach of using microdialysis techniques has been used extensively to assess changes in neurotransmitter responsiveness following a variety of stressors (environmental, age, pharmacological).

See response above regarding sentence 233. This text has been modified to accurately reflect pairwise interactions between neurotransmitters based on values that only include Pearson's correlation ≥ 0.5 with a significance p-value of <0.05 (see also modified supplementary Table 2).

6c) Applications of high k⁺ concentrations (100 mM) have been shown to induce distinct oscillations in DA levels. I am wondering why the authors increased the K⁺ stimulation to 120mM but didn't discuss the consequences of such stimulation and that this has not much to do with behavioral patterns under normal conditions. Simply arguing that "the temporal patterns of other neurotransmitters were less affected, requiring greater local K⁺-362 stimulation to observe any differences", is not sufficient. Very high stimulations lead to very different dynamics, it is not necessarily linear.

As noted in other responses (reviewer 2, points 3 and 6biii and reviewer 3, point 5) the low to high K⁺ concentrations were selected not to reproduce functionally relevant physiological conditions, but to perturb the system in a manner that may elucidate more subtle changes that would not be revealed at lower levels of K⁺-stimulation as well as more obvious alterations in the neurotransmitter network dynamics that are apparent at high K⁺ stimulation. This point raised by the reviewer of differing neurotransmitter dynamics at different levels of stimulation is precisely the reason why we used a powerful approach of documenting a K⁺-evoked concentration dependent (i.e., a dose-response curve) change in neurotransmitter responsiveness. Interestingly, our Pearson's correlation and machine learning network analysis as well as the new Granger Causality test provides clear evidence for this K⁺-evoked concentration-dependent dynamic changes in neurotransmitter networks. In the revised manuscript, we have added text to clarify this.

Please note that high K⁺-stimulation (100-120 mM) is commonly used within the literature to assess changes in brain neurochemistry, as demonstrated by the sample publications below (see also response to reviewer 2, point 3 and point 5 under reviewer 3). Text has been added to page 49 (Online methods section) to further clarify the rationale for using low to high K⁺.

1. Maidment, N. T., Brumbaugh, D. R., Rudolph, V. D., Erdelyi, E., & Evans, C. J. (1989). Microdialysis of extracellular endogenous opioid peptides from rat brain in vivo. *Neuroscience*, 33(3), 549–557. [https://doi.org/10.1016/0306-4522\(89\)90407-7](https://doi.org/10.1016/0306-4522(89)90407-7)
2. Ngo, K. T., Varner, E. L., Michael, A. C., & Weber, S. G. (2017). Monitoring Dopamine Responses to Potassium Ion and Nomifensine by in Vivo Microdialysis with Online Liquid Chromatography at One-Minute Resolution. *ACS Chemical Neuroscience*, 8(2), 329–338. <https://doi.org/10.1021/acschemneuro.6b00383>
3. Ripley, T. L., Jaworski, J., Randall, P. K., & Gonzales, R. A. (1997). Repeated perfusion with elevated potassium in in vivo microdialysis—A method for detecting small changes in extracellular dopamine. *Journal of Neuroscience Methods*, 78(1), 7–14. [https://doi.org/10.1016/S0165-0270\(97\)00129-5](https://doi.org/10.1016/S0165-0270(97)00129-5)
4. Yang, H., Sampson, M. M., Senturk, D., & Andrews, A. M. (2015). Sex- and SERT-Mediated Differences in Stimulated Serotonin Revealed by Fast Microdialysis. *ACS Chemical Neuroscience*, 6(8), 1487–1501. <https://doi.org/10.1021/acschemneuro.5b00132>

7) Page 15 line 316: The authors state "Likewise, we found that acute or chronic delivery of the 33-beam GCR generally produced similar and less severe neurocognitive and electrophysiological impairments in rodents". In this paper, there is not a single electrophysiological experiment or even statement. Even if the authors were partly involved in the other study [34], this sentence needs to be rephrased. Otherwise, it is very misleading.

Thank you for pointing this out. The foregoing sentence has now been reworded (see page 15-16): In previous work, cohorts of male and female mice exposed to the same acute or chronic delivery of the 33-beam GCR were found to exhibit similar and less severe neurocognitive and electrophysiological impairments³⁴.

8) Page 17 line 354: The authors mention that the results with respect to glu-gaba system, which is the most critical system within cortical structures, is not in agreement with previous studies. How do the authors justify this?

We apologize for any confusion. This statement is mainly related to the fact that our data with the complex GCR exposure on baseline levels of glutamate and GABA are somewhat different than previous work that examined effects of simplified radiation exposures on these systems. In the revised manuscript we have now clarified this further (see page 18). From a more border perspective, previous studies have used simplified single ion exposures which may not produce the physiologically relevant changes desired to simulate deep space radiation. By using a complex 33-beam GCR simulation we more accurately reproduce the physiologically relevant conditions, which may produce differing results from simpler, less representative models. However, we emphasize that much our past work using complex GCR (Alaghband et al., 2023; doi: 10.1007/s00018-022-04666-8), mixed field 5 ion studies (Klein et al., 2021; doi: 10.1016/j.nbd.2021.105252; Keiser et al., 2021; doi: 10.1016/j.nlm.2020.107367) and even acute (Klein et al., 2021; doi: 10.3390/ijms22169020) and chronic low dose neutron (Acharya et al., 2019; doi: 10.1523/ENEURO.0094-19) studies suggest multiple disruptions across interconnected brain regions. These extensive data sets are consistent with our arguments highlighted by causal machine learning analyses pointing to a high probability that the current and even past exposure paradigms likely elicit significant reorganization of neurotransmitter networks. One way to envision such changes is that a relatively small dose can trigger a chain reaction of events leading to the accumulation of many subtle changes that individually may not be consequential (evidenced by the relatively minor impact of space radiation exposure on paired cell recordings found in the aforementioned references) but collectively lead to the types of global alterations in network connectivity reported here.

9) Page 18 line 384: This interpretation is far over-fetched and not really clear. The corr coeffs are so low that there is not real meaning in it. Moreover, it is not clear why EIN is lost in baseline case.

Please refer to response to reviewer 2, point 6a, 6bi, and 6ii. The revised network plots (Fig. 7c) and analysis based on values that only include Pearson's correlation ≥ 0.5 with a significance p-value of <0.05 (see also modified supplementary Table 2) further confirm our overall interpretation that acute and chronic GCR significantly reorganize the neurotransmitter networks in the PFC. As mentioned above, the figures and results section has been modified to accurately reflect this point (see also modified supplementary Table 2).

10) Page 36 online methods: There are certain standards of reporting methods that are not followed. number of animals is not provided. Origin of the animals is not clear. Currently, it is a copy-paste ("Jackson Laboratories, City, State"). The statement about "-6-9 months old (-20-25 years in human age)" is not meaningful. It is not discussed why no female animals were used although many studies indicate sex-specific effects.

These oversights have been addressed. The location of the Jackson Laboratory has been updated. The age of the animals and the relationship to human aging has been further elaborated on to better convey the meaning of the statement. See response to reviewer 3, point 3 regarding the lack of female animals in these studies.

11) In summary, the topic of this study is interesting but in my humble opinion, the choice of methods and the findings do not fully justify the main conclusion of this study.

We respectfully disagree. In this manuscript, we have tried to emphasize the novelty of a completely different within-subject approach for extracting and analyzing neurochemical data in freely moving animals (rather than *in vitro* tissue content analysis) that may well inform on the types of long-term behavioral and neurocognitive changes that can be important for mission success during deep space travel. The combination of select translational touchscreen behavioral assessments coupled with *state-of-the-art in vivo* neurotransmitter analyses using LC-MS/MS provides a unique snapshot into the key biochemical alterations that persist in the GCR exposed rodent brain. Such a within-subject approach is novel and has not been previously utilized in space radiobiology studies. Lastly, we tied everything together with extensive correlation and machine learning approaches, including the Granger Causality analysis, all of which have uncovered some fascinating insights on how GCR exposures might elicit functional reorganization of neurotransmitter networks in the PFC, especially regarding the monoamines (see above responses to 6a and 6bii). While we agree that research questions remain, we posit that our approach provides a novel conceptual framework to rationalize much of what has been reported in the field. We did our best to encapsulate this in our revised discussion.

Reviewer #3:

1) Desai and collaborators (ms#NCOMMS-23-06284-T) study a fascinating topic regarding the potential impact of galactic cosmic rays (GCR) exposure during space exploration in cognitive behaviors and neurochemistry. For this purpose, authors simulated GCR in the lab by acutely or chronically exposing adult C57BL/6 mice to a complex 33-beam composition of GCR. Observations were compared to control mice not exposed to GCR but otherwise underwent similar handling and experimental procedures. GCR exposed mice were then behaviorally tested using automated touchscreens. Neither acute nor chronic GRC exposure affected reward sensitivity during an Economic Demand Task but increased reaction times to respond to stimuli in a Psychomotor Vigilance Task. Finally, in vivo microdialysis experiments recorded extracellular baseline and KCl-evoked levels of dopamine, GABA, glutamate, serotonin (5-HT), and norepinephrine in the prefrontal cortex (PFC). Authors report that acute and/or chronic exposure to GCR increased basal and KCl-evoked levels of all the neurotransmitters tested when compared to levels in control mice. Finally, authors used a correlational and machine learning approach to study whether changes in the GCR-induced disbalance in PFC impacted neurotransmitter networks.

Thank you for these positive comments, we tried to present a series of very novel experiments that provide deeper insight into the problems associated with radiation exposure of the brain during deep space travel.

Major Comments:

2) Overall, experiments and data presented here seem well performed. The major criticism is however that this manuscript lacks a rationale/story line that helps the reader to understand the selection of the experiments performed, and why they are relevant in the conclusions here depicted. For instance, it is not fully clear why a behavioral task to assess reward sensitivity was used along an attentional task if the main scope was to test for potential cognitive deficits related to exposure of GCR. Was this the only goal in this study? Furthermore, why authors decided to test for neurotransmitters in the PFC alone, but not from other brain regions also directly or indirectly associated to cognitive function? What was the rationale to study only the above-mentioned neurotransmitters and not others? Why is relevant a correlation analysis between pair of neurotransmitters? Did the authors consider any functional criteria (e.g., biosynthetic pathways, excitability, neuromodulatory, etc.) to match and assign different weights of functional relevance? How this analysis helps to understand neurochemical networks in the PFC?

We thank the reviewer for the kind comments, constructive criticism, and opportunity to respond/revise. Given the high value of these subjects following GCR exposure, we designed these studies using a within-subject design to derive multiple behavioral and neurochemical endpoints in each mouse to allow for correlational analysis of potential brain/behavior deficits. We chose these particular tasks because we (and our NASA advisors) believe that employing animal models of reward sensitivity (economic demand) and focused attention (psychomotor vigilance) capture critical features of mission-critical performance. Specifically, although we agree with the reviewer that reward sensitivity via economic demand indices is not particularly “cognitive”, they can negatively affect virtually all aspects of cognition-related behavior if a subject is blunted to appetitive reward. Regarding the psychomotor vigilance task, we think the attentional processes required to develop and maintain focused vigilance are (albeit simplified) elements of cognitive performance. Moreover, the GCR-related deficits observed in these studies are likely to be highly relevant to the various duties an astronaut will need to perform throughout extended spaceflight. They are also tasks that can be directly linked to neurotransmitter changes in key brain regions of interest for the above-mentioned correlational and machine learning analyses. See also responses to reviewer 3, points 3–5.

The reviewer raises an important point regarding testing neurotransmitters in the PFC alone. As stated above (see also our response to reviewer 2, point 2), while it would be highly exciting to obtain information on the neurotransmitter dynamics from multiple brain regions to document how, when, and where GCR exposure is altering brain neurotransmitters (e.g., a “neurochemical connectome”) to impact behavior/cognition, such an endeavor would require a substantially larger cohort of animals that were simply not available for this study. Consequently, based on well-established reports, the PFC was selected because of the extensive body of evidence showing its involvement in higher order behaviors and cognitive function, including motivation and psychomotor vigilance that are likely required for mission success during spaceflight. Below, we provide a sample of publications highlighting the relevance of the PFC to the behaviors studied here. Regarding the selection of neurotransmitters, we primarily focused on this group of key neurotransmitters in the PFC because prior work has shown a) that they all play an important role in various cognitive processes, including motivation and attention and b) that their dysregulation is directly linked to a variety of neuropsychiatric conditions. While it is possible to detect other neurochemicals using our LC-MS approach, we believe that the data presented here sets a strong benchmark for an expanded evaluation of the neurochemical profile not only of PFC but also other key brain regions. In the revised manuscript, we have added text in the introduction to provide a better rationale for studying the PFC and these neurotransmitters on page 6.

1. Arnsten, A. F. T., & Rubia, K. (2012). Neurobiological Circuits Regulating Attention, Cognitive Control, Motivation, and Emotion: Disruptions in Neurodevelopmental Psychiatric Disorders. *Journal of the American Academy of Child & Adolescent Psychiatry*, 51(4), 356–367. <https://doi.org/10.1016/j.jaac.2012.01.008>

2. Arnsten, A. F. T., Wang, M., & Paspalas, C. D. (2015). Dopamine's Actions in Primate Prefrontal Cortex: Challenges for Treating Cognitive Disorders. *Pharmacological Reviews*, 67(3), 681–696. <https://doi.org/10.1124/pr.115.010512>
3. Bahmani, Z., Clark, K., Merrikhi, Y., Mueller, A., Pettine, W., Isabel Vanegas, M., Moore, T., & Noudoost, B. (2019). Prefrontal Contributions to Attention and Working Memory. In T. Hodgson (Ed.), *Processes of Visuospatial Attention and Working Memory* (pp. 129–153). Springer International Publishing. https://doi.org/10.1007/7854_2018_74
4. Rossi, A. F., Pessoa, L., Desimone, R., & Ungerleider, L. G. (2009). The prefrontal cortex and the executive control of attention. *Experimental Brain Research. Experimentelle Hirnforschung. Experimentation Cerebrale*, 192(3), 489–497. <https://doi.org/10.1007/s00221-008-1642-z>
5. Siddiqui, S. V., Chatterjee, U., Kumar, D., Siddiqui, A., & Goyal, N. (2008). Neuropsychology of prefrontal cortex. *Indian Journal of Psychiatry*, 50(3), 202–208. <https://doi.org/10.4103/0019-5545.43634>

Regarding relevance of correlation analysis please see responses to reviewer 2, point 6a–c.

3) Additionally, it is hard to connect the significance of the different datasets presented in this manuscript into a single major conclusion. For example, why GCR affected behavior and extracellular neurotransmitters in the PFC? what is the relationship between an increased reaction time to touch stimuli in GCR-exposed mice and the abnormal neurochemical fingerprint in PFC? What do we learn by using a KCl-evoked response that cannot be concluded from the baseline levels alone? How altered neurochemical networks in the PFC may explain an abnormal behavioral phenotype?

The reviewer raises some interesting points, many of which we have pondered extensively during the interpretation of our datasets. Investigators engaged in this field have struggled with identifying definitive mechanisms able to account for adverse functional outcomes such as those reported here for the PVT. From a global perspective, the entire brain is susceptible to radiation injury, but clearly will not exhibit equivalent levels of sensitivity. Behavior is a multifaceted response to a variety of molecular and biochemical changes impacting neurotransmission, and GCR exposure (and many past irradiation paradigms) have been shown to produce persistent changes in functional and molecular outcomes in the CNS. Past studies suggested that such exposures might also elicit changes in neurotransmitter availability and/or alter the dynamic response of neurotransmitters available which can greatly impact neurocognitive function. Increased levels of neurotransmitters (Figs. 5 and 6) may well predispose certain excitatory and inhibitory circuits to fire asynchronously, which may not lead to overt changes in behavior under basal operating conditions, but under situations of exaggerated spaceflight stress, critical, accurate and timely decision making may be compromised. For example, if we translate increasing K⁺ levels to increasing levels of spaceflight stress then muted dynamic neurotransmitter responses (particularly with DA, Fig. 6a) could easily compromise reaction times and adaptive decision making – a central role for the PFC. As pointed out in our discussion, these changes are not likely restricted to the PFC, but reflect a mere snapshot of the types of alterations likely residing in other regions of the brain important for executive function. While we cannot pinpoint a precise adverse behavior associated with the neurotransmitter findings, our data showing neurotransmitter network reorganization are equally hard to dismiss, and likely represent some of the fundamental biochemical changes that are contributory if not causal to space radiation-induced behavioral and neurocognitive decrements. In this regard, our systematic approach of using correlation and machine learning analysis, including the new Granger Causality test uncovered some fascinating insights on how GCR exposures might elicit functional reorganization of neurotransmitter networks in the PFC, especially regarding the monoamines. As stated above Figure 9 confirms significant changes in the response of the PFC neurotransmitter network depending on the GCR exposure (see also response to 6a and 6bii). Portions of the foregoing have been added to the discussion at your request (see pages 18, 19, 21, 22).

4) Economic Demand task: Unclear why authors are comparing performance following different

reward concentrations (50%, 20%, 5% or 0% milk concentration) and different responding, all throughout the same task. Is there evidence that this is a measure of motivation? More discussion of the rationale for using this task and the dependency of it on the PFC is needed.

We thank the reviewer for the opportunity to clarify. It is a common approach to vary reward magnitude in economic demand procedures to evaluate a subject's *sensitivity to reward*. Likewise, it is also common to vary the response requirement for reward to evaluate a subject's *motivation for reward*. Although we and others have documented adverse effects of environmental stressors or drug treatment that negatively affect sensitivity and/or motivation, it is interesting and important to document this null effect following GCR exposure. Incidentally, this null effect also provides evidence that the PVT deficits documented in the subsequent study were not simply a consequence of a GCR-induced deficit in reward valuation but rather attentional processes. See response to reviewer 3, point 3 above regarding its dependency on the PFC. As the reviewer recommends, we now add discussion on these points on page 14-15.

5) Psychomotor Vigilance Task: Why choose 20% concentration for this task following the variable reward concentrations of previous task? Explain. Acute and chronic exposure increased response times. Is this a real attentional impairment? Is there evidence that this not a learning impairment (as the reaction times are most effected early in training) or a motor impairment?

We have examined a variety of milk concentrations in the maintenance of touchscreen-related behavior in the mouse and have determined that 20% is a highly effective reward magnitude for maintaining such performance across numerous trials without risk of satiation (see Hisey EE, Fritsch EL, Newman EL, Ressler KJ, Kangas BD, Carlezon WA Jr. Early life stress in male mice blunts responsiveness in a translationally-relevant reward task. *Neuropsychopharmacology*. 2023 May 31. doi: 10.1038/s41386-023-01610-7. Epub ahead of print. PMID: 37258714). We do indeed believe that in subjects following either acute or chronic GCR exposure, true deficits in attentional processes were observed. As we discuss, inferior performance in both the acquisition and subsequent steady state reaction time metrics provides strong evidence of this fact. It is conceivable that GCR-related motoric abnormalities may have contributed, but our findings, in the same subjects, under the economic demand studies makes this possibility unlikely, especially considering the high response requirements in those conditions.

Minor Comments:

6) Abstract: why authors mention only dopamine but not the other neurotransmitters tested?

We focused on dopamine changes in the abstract as our data clearly showed that it is a key neurotransmitter that is being altered and is influencing other neurotransmitter network dynamics. The abolished temporal DA responsiveness following GCR exposure vs controls is especially striking and is in line with the extensive body of evidence suggesting PFC DA as a mediator of attention. As such discussing DA-related changes in the context of also discussing the results of the attention-related behavioral assays seemed appropriate. Please note that we do refer to other neurotransmitters in lines 48 and 49 to indicate that while dopamine is the headlining neurotransmitter, multiple others were also analyzed in this study. Mention of the relevant neurochemistry is addressed in the same statement used to address author response to review 3 major comment 3.

Practically, the 150-word limit for abstracts also precludes us from providing additional details on individual neurotransmitter changes.

7) Introduction: The study of PFC circuits – it is mentioned in the introduction that neurocognitive performance should be studied, but the rationale for studying PFC is lacking.

See response to reviewer 3, major comment 3 and 5 regarding the importance of the PFC for the behavioral assays used in the manuscript. In the revised manuscript, the text in the introduction has been amended to clearly highlight the key role that PFC plays in reward sensitivity and attentional processes.

8) Must be more clear what sexes you are studying (only males) and why. In the discussions, authors specify the role of sex in similar work by Dr. Amelia Eisch, but do not discuss this further in relations to their own findings.

The male mice were made available from a larger cohort of both sexes subjected to the acute and chronic GCR simulation (see Alaghband et al., 2023, PMID: 36607431). Female mice were simply not available for these studies. Regarding differential responses between the sexes exposed to GCR, work published from a similarly exposed cohort found female mice somewhat more resistant to cognitive deficits (Alaghband et al., 2023, PMID: 36607431), which paralleled past results obtained with single Helium ion exposures (Parihar et al., 2020, PMID: 33192361). While the foregoing is less relevant to the current study since only male mice were available for study, our group has suggested that higher baseline levels of inflammation may precondition the female rodent brain, thereby attenuating the extent of space radiation injury and possibly ameliorating resultant cognitive impairments. In the revised manuscript, we have now included text that further discusses this issue. We have included certain elements of the foregoing in the discussion at your request (see page 15).

9) Age of mice: on line 726: specifies that they used 6-9 months of age mice which is equivalent to 20-25 years of human age. According to Jackson laboratory, 6-9 months of age would be equivalent to 30-40 years of human age

The reviewer is correct. Thank you for pointing this out. This has been amended in the text.

10) Graphs should have SEM or individual variability presented for readers interpretation of results (Fig 2c).

We thank the reviewer for this excellent suggestion. We have now revised Fig 2c and added SEM to all results.

11) Line 43: Typo ‘prefrontal’

This has been corrected.

12) Line 170: Typo ‘Neurotransmitters’

This has been corrected.

13) Lines 195, 197, 204, 211, 214, 217, 219: please provide units to data presented

In the revised manuscript, we now include units in figure 6 and the figure legend as % of basal (nM) levels. Units have also been included in the text where appropriate (see page 9-11). Please note that dialysis data are typically normalized to basal levels and are reported accordingly.

14) Lines 734-745 (irradiation protocol): Could you please comment why acute vs. chronic exposure radiation used different times (2 hours vs. 80 minutes)? Also, why animals were restrained ≈20 mins (line 745) if radiation times were longer?

Irradiation procedures used in this study were precisely the same as those described in our recent publication (Alaghband et al., 2023, PMID 36607431, reference 34). Acutely irradiated animals received a GCR simulation dose of ~40cGy over the duration of 2h on the same day. Chronically irradiated animals received of 2.08 cGy/day, 6 days a week for 4 weeks for a total accumulated dose of ~50 cGy that took approximately 1h to complete/day. Over the duration of exposure animals were confined to a small, ventilated plastic box. Irradiation times were fixed by the technical limitations at the NASA’s Space Radiation Laboratory (Brookhaven National Laboratory) of delivering this complex 33-beam GCR simulation exposure. The text has been amended in the Methods section to clarify these details on page 39 (see Online Methods section).

15) Line 850: What was the molecular weight cut-off of your dialysis probe?

The molecular weight cutoff for our dialysis AN69 fibers is between 10-20 kDa. This detail is now included in the Methods on page 45 (see Online Methods section). Note that the molecular weight cut-off for the

detected neurotransmitters in this manuscript as well as typical neurotransmitters and their metabolites is usually less than 1 kDa.

16) Lines 867-869: why did you use 3 different concentrations of KCl?

See responses to reviewer 2, points 3 and 6biii and reviewer 3, point 5.

17) Figure 4c: please provide specific probe location instead of overall region. As suggested by the reviewer, the revised Figure 4c displays the precise location of each probe within the

PFC.

18) Figure 7b and 8b: figures seem stretched in the vertical plane. Please correct. This has been corrected in the revised figure.

REVIEWERS' COMMENTS

Reviewer #1 (Remarks to the Author):

The authors have addressed most of my comments from the previous revision. However, I continue to disagree with some of the over interpretation of the behavioral data. First, while they acknowledge that the differences are “development” of the attention task – which is simply acquisition of this task – they continue to argue for differences in asymptotic performances that are not supported by their data. From my reading of the paper, the average reaction time values only differ from control for both groups until day 5, and from the acute group on day 6, but at day 7 and beyond there are no significant differences between the groups. The authors now state on page 16, lines 335-342 that asymptotic performances never approximated controls, but this is merely descriptive and there is no statistical analysis to support this conclusion (i.e., the irradiated groups do not remain statistically different from controls). Further, the assignment of asymptotic performance is also descriptive; there is no statistical analysis to show when each group differed from its initial reaction time value or when its day-to-day performance did not differ/was stable. While I understand that they are defining this as the fact that reaction time values no longer decrease at a certain day, some formal analysis is needed to support the argument that asymptotic values differ following radiation exposure.

Reviewer #2 (Remarks to the Author):

I wish to thank the authors to carefully pay attention to my comments and concerns. In my opinion, the authors did all possible efforts to address my concerns. I am aware that to address some of my comments, they might have needed a year to conduct additional studies.

While I still believe that this paper could become a truly seminal study with far less weaknesses, I also think that the authors have now improved it sufficiently so that an expert reader can identify critical issues and potential shortcomings and treat certain conclusions with more care.

I just would like to ask the authors to note that at high levels K+ stimulations, there are major induced oscillations in the glu-gaba networks. This will have a non-negligible dynamical effects on their findings. I have been aware of all the papers they refer to. That other studies use similar doses to perturb the system, doesnt mean that one should ignore what type of perturbations one induced. In my humble opinion, it would make sense to not ignore this.

Reviewer #3 (Remarks to the Author):

The authors addressed all my comments and am happy with the outcome. Congratulations and best wishes.

Response To Reviewers:

Reviewer #1:

1) The authors have addressed most of my comments from the previous revision. However, I continue to disagree with some of the over interpretation of the behavioral data. First, while they acknowledge that the differences are “development” of the attention task – which is simply acquisition of this task – they continue to argue for differences in asymptotic performances that are not supported by their data. From my reading of the paper, the average reaction time values only differ from control for both groups until day 5, and from the acute group on day 6, but at day 7 and beyond there are no significant differences between the groups. The authors now state on page 16, lines 335-342 that asymptotic performances never approximated controls, but this is merely descriptive and there is no statistical analysis to support this conclusion (i.e., the irradiated groups do not remain statistically different from controls). Further, the assignment of asymptotic performance is also descriptive; there is no statistical analysis to show when each group differed from its initial reaction time value or when its day-to-day performance did not differ/was stable. While I understand that they are defining this as the fact that reaction time values no longer decrease at a certain day, some formal analysis is needed to support the argument that asymptotic values differ following radiation exposure.

We are delighted to hear that we satisfied almost all of Reviewer 1’s previous concerns. And we appreciate Reviewer 1’s point about the descriptive nature of our discussion related to GCR treatment groups’ failure to fully approximate reaction times of the average control subject. The reviewer is correct that this portion of the function was not statistically significant. As such, we have now revised this text to make clear that, although performance in the acute and chronic GCR groups failed to fully approximate asymptotic levels observed in control subjects by ~1 sec, and that we believe these small differences could be meaningful for the astronaut experience, these differences did not meet statistical criteria. Please see page 8/9 and 16 for these changes (highlighted in red).

Reviewer #2:

1) I wish to thank the authors to carefully pay attention to my comments and concerns. In my opinion, the authors did all possible efforts to address my concerns. I am aware that to address some of my comments, they might have needed a year to conduct additional studies. While I still believe that this paper could become a truly seminal study with far less weaknesses, I also think that the authors have now improved it sufficiently so that an expert reader can identify critical issues and potential shortcomings and treat certain conclusions with more care.

We thank the reviewer for their thoughtful comments and are delighted to hear that we have now addressed most of their comments and concerns.

2) I just would like to ask the authors to note that at high levels K+ stimulations, there are major induced oscillations in the glu-gaba networks. This will have a non-negligible dynamical effects on their findings. I have been aware of all the papers they refer to. That other studies use similar doses to perturb the system, doesn’t mean that one should ignore what type of perturbations one induced. In my humble opinion, it would make sense to not ignore this.

The reviewer’s point is well-taken. Accordingly, we have now included text on page 23 in the discussion to highlight that high levels of K+ stimulations can also induce changes in neurotransmitters efflux and may confound current interpretations.

Reviewer #3:

1) The authors addressed all my comments and am happy with the outcome. Congratulations and best wishes.

Thank you!